# Hygroscopic growth study in the framework of EARLINET during the SLOPE I campaign: synergy of remote sensing and in-situ instrumentation

Andrés E. Bedoya-Velásquez [1,2,3], Francisco Navas-Guzmán [4], María J. Granados-Muñoz [5], Gloria Titos [1,6], Roberto Román[1,2,11], Juan A. Casquero-Vera [1,2], Pablo Ortiz-Amezcua [1,2], Jose A. Benavent-Oltra [1,2], Gregori de Arruda Moreira[1,2,7], Elena Montilla-Rosero [8], Carlos D. Hoyos [9], Begoña Artiñano [10], Esther Coz [10], Lucas Alados-Arboledas [1,2] and Juan Luis Guerrero- Rascado [1,2].

[1]Andalusian Institute for Earth System Research (IISTA-CEAMA), University of Granada, Autonomous Government of Andalusia. 18006, Granada, Spain.
[2]Departament of Applied Physics, University of Granada. Granada, Spain.
[3]Sciences Faculty, Department of Physics, Universidad Nacional de Colombia. Medellín, Colombia.
[4]Federal Office of Meteorology and Climatology MeteoSwiss, Payerne, Switzerland.
[5]Remote Sensing Laboratory / CommSensLab, Universitat Politècnica de Catalunya, Barcelona, 08034, Spain.
[6]Institute of Environmental Assessment and Water Research (IDAEA), CSIC, Barcelona, 08034, Spain.
[7]Institute of Research and Nuclear Energy, IPEN.São Paulo, Brasil.
[8]Physical Sciences Department, School of Science,EAFIT University, Medellín, Colombia.
[9]Minas Faculty, Department of Geosciences and Environment, Universidad Nacional de Colombia.Medellín, Colombia.
[10]CIEMAT, Environment Department, Associated Unit to CSIC on Atmospheric Pollution, Avenida Complutense 40, Madrid, Spain.
[11]Grupo de Óptica Atmosférica (GOA), Universidad de Valladolid. Paseo Belén, 7, 47011, Valladolid, Spain.

*Correspondence to*: Andrés Esteban Bedoya Velásquez (aebedoyav@correo.ugr.es)

## Abstract

This study focuses on the study of aerosol hygroscopic growth during Sierra Nevada Lidar AerOsol Profiling Experiment (SLOPE I) campaign by using the synergy of active and passive remote sensors at Granada valley station and in-situ instrumentation at a mountain station (Sierra Nevada). To this end, a methodology based on the combination of calibrated water vapour mixing ratio (r) profiles, retrieved from an EARLINET multiwavelength Raman lidar (RL), and continuous temperature profiles from a microwave radiometer (MWR) for obtaining relative humidity (RH) profiles with high temporal resolution is used. This methodology is validated against an approach using radiosounding (RS) data, obtaining differences in hygroscopic growth parameter (γ) lower than 5% between the methodology based on RS and that based on remote sensing. During SLOPE I the remote sensing methodology used for aerosol hygroscopic growth studies has been checked against Mie calculations of hygroscopic growth using in-situ measurements of particle number size distribution measured at SNS. The hygroscopic case observed during SLOPE I showed an increase in particle backscatter coefficient at 355 and 532 nm with the relative humidity (with RH ranging between 78-98%), but also a decrease in backscatter-related Ångström exponent (AE) and

particle linear depolarization ratio (PLDR) indicating that the particle became larger and more spherical due to hygroscopic processes. Vertical and horizontal wind analysis is performed by means of a co-located Doppler lidar system at IISTA-CEAMA station, in order to evaluate the horizontal and vertical dynamics of the air masses. Finally, the Hänel parameterization is applied to experimental data for both stations and we found good agreement on $\gamma$ parameters measured ($\gamma^{532} = 0.48 \pm 0.01$

and $\gamma^{355} = 0.40 \pm 0.01$) respect to calculated ($\gamma^{532} = 0.53 \pm 0.02$ and $\gamma^{355} = 0.45 \pm 0.02$), with relative differences between measured and calculated up to 9 % at 532 nm and 11 % at 355 nm.

KEYWORDS: ACTRIS, Aerosol hygroscopic growth, Doppler lidar, EARLINET, lidar, microwave radiometry, remote sensing.

**1 Introduction**

Atmospheric aerosol particles play a crucial role in the Earth´s climate, principally by means of the radiative effect due to aerosol-radiation and aerosol-cloud interactions, affecting the Earth-atmosphere energy balance and, hence, the Earth's climate. Furthermore, the aerosol might also modify optical and microphysical cloud properties, such as albedo and cloud

droplet size distribution that influences cloud lifetime, since the particles could act as cloud condensation nuclei (CCN) and ice nuclei (IN) (Twomey, 1977; Albrecht, 1989; IPCC, 2013).

Water vapor plays a major role in the aerosol-radiation interaction due to the ability of some atmospheric aerosol particles to take up water from the environment. In this sense, hygroscopic growth is the process by which aerosol particles uptake water

and increase their size under high relative humidity (RH) conditions (Hänel, 1976). Consequently, this process is also related to changes in the optical and microphysical properties of the aerosol particles and, hence, it becomes a crucial factor that modifies the role of aerosols in atmospheric processes and radiative forcing.

Several studies have been carried out over the past years in order to evaluate how water uptake affects aerosol properties. One

parameter used to quantify these changes is the so-called aerosol hygroscopic enhancement factor, f ($\lambda$,RH), defined as the ratio between aerosol optical/microphysical properties at wet atmospheric conditions and the corresponding reference value at dry conditions (Hänel 1976; Ferrare et al. 1998; Feingold et al., 2003; Veselovskii et al., 2009; Granados-Muñoz et al., 2015; Titos et al., 2014, 2016, and references therein). Most of the previous studies investigating aerosol hygroscopicity are based on in-situ measurements. One of the most commonly used in-situ instruments for measuring aerosol hygroscopicity is the

Humidified Tandem Differential Mobility Analyzer (HTDMA) (e.g. Swietlicki et al., 2008) that measures the hygroscopic growth factor g (RH) that quantifies the change in particle diameter due to water uptake. Humidified tandem nephelometers have been extensively used as well to quantify the effect of the hygroscopic growth in the aerosol optical properties, namely

scattering, backscattering and extinction coefficients (e.g., Pilat and Charlson 1966; Titos et al., 2016). There are other in-situ instruments such as the white-light humidified optical particle spectrometer (WHOPS) (Rosatti et al., 2015) or the Differential Aerosol Sizing and Hygroscopicity Spectrometer Probe (DASH-SP), (Sorooshian et al., 2008) that have been used to determine the impact of enhance RH on the aerosol properties from airborne platforms. The effect of RH on the aerosol optical properties

can be also determined with Mie model calculations (e.g. Adams et al., 2012; Fierz-Schmidhauser et al., 2010; Zieger et al., 2013) using the measured size distribution and chemical composition as inputs. For this calculation, the g (RH) is also needed as input. This factor can be determined experimentally (using HTDMA measurements for example) or it can be inferred from the individual growth factors of the different chemical compounds. The assumption of some aerosol properties such as the refractive index or the growth factor based on the chemical composition is the main drawback of this method. In general terms,

one important limitation of most in-situ techniques is that they modify the ambient conditions and are also subject of particles losses in the sampling lines, therefore altering the real atmospheric aerosol properties.

Remote sensing systems such as lidars have also been used in the last decades for aerosol hygroscopic growth studies performed with co-located RS measurements (e. g. Ferrare et al., 1998; Feingold et al., 2003; Veselovskii et al., 2009;

Granados-Muñoz et al., 2015; Fernández et al., 2015; Lv et al., 2017). These systems have shown to be robust with high vertical and temporal resolution that allow for studying the aerosol hygroscopic growth under unmodified ambient conditions. This aim has also been studied by Zieger et at. (2011), showing the capability to combine Raman lidar, in-situ and MAX-DOAS instrumentation for study hygroscopic growth in ambient conditions extrapolating extinction coefficient from lidar to the ground studies. Also, some studies have been performed by using Automatic Lidar and Ceilometers (ALC) for hygroscopic

and fog studies mostly for forecasting purposes of fog events, through the combination of attenuated backscatter with in-situ data from instrumented towers which reach almost 200 m above ground level (Haeffelin et al., 2016). In this work we are adding a comparison between ground city station to high mountain station in order to connect effects of the city over mountain and also avoid technical issues like lidar overlap.

Up to now, most hygroscopic growth using lidar systems combine the lidar measurements with RH data from RS. The main inconveniences are that RS measurements have low temporal sampling and they could be drifted away from the vertical atmosphere probed by the lidar systems.  These inconveniences can be easily overcome by combining calibrated water vapor mixing ratio profiles, r (z), from Raman lidar (RL), with temperature profiles from ancillary instrumentation for obtaining RH and aerosol backscatter profiles, using them simultaneously for hygroscopic growth studies (e.g. Whiteman, 2003; Navas-

Guzmán et al., 2014; Barrera-Verdejo et al., 2016). Navas-Guzmán et al. (2014) proposed a methodology for retrieving RH profiles by the combination of calibrated r (z) profiles from Raman lidar water vapor channel with temperature profiles obtained from microwave radiometer (MWR) measurements. RH profiles obtained using this approach and aerosol profiles from the lidar are used in this work to study aerosol hygroscopic growth. This methodology allows for obtaining a wider database for the analysis of the aerosol hygroscopic growth properties using remote sensors since some of the limitations

associated to RS are overcome. Additionally, water vapour and aerosol measurements are performed with the same system and, thus, the same air volume is probed, avoiding the radiosonde drift and temporal mismatching sampling.

The main goal of this study is to apply the methodology proposed by Navas-Guzmán et al. (2014), based on the application of the synergy between RL and MWR for aerosol hygroscopic growth studies. First, this methodology for hygroscopic growth studies is compared with the approach presented in Granados-Muñoz et al. (2015) that uses RS and lidar data. Once the technique is validated, a study of the aerosol hygroscopic growth case observed during the SLOPE I (the Sierra Nevada Lidar AerOsol Profiling Experiment I) campaign is presented. The results obtained with remote sensing are compared with Mie simulations performed using in-situ measurements from a high-mountain station (up to 2500 m a.s.l).

This paper is organized as follows. Description of the experimental site and instrumentation are presented in Section 2. The methodology applied is introduced in Section 3. Section 4 presents the results and discussion of the combination of RL and MWR method for obtaining RH profiles and also the hygroscopic case analysed by combining measured and retrieved data from lidar and in-situ instrumentation, respectively. Finally, conclusions are given in Section 5.

## 2 Experimental site and instrumentation

### 2.1 SLOPE I field campaign

In summer 2016, the Sierra Nevada Lidar AerOsol Profiling Experiment (SLOPE I) intensive field campaign was carried out in South-Eastern Spain in the framework of the European infrastructure ACTRIS. The goal of this campaign was to perform a closure study by comparing remote sensing and in-situ measurements at different altitudes taking advantage of a unique experimental setup. This setup consisted of several experimental stations located at different altitude levels in the slope of Sierra Nevada, which is located 20 km away in horizontal distance from the remote sensors at IISTA-CEAMA station. In this work, we only use the station located at 2500 m a.s.l, which has the in-situ instrumentation needed (see on Fig. 1). Therefore, combined active and passive remote sensing measurements using multiple instrumentation at the IISTA-CEAMA station and simultaneous in-situ measurements at 2500 m a.s.l in the northern slope of Sierra Nevada were performed from May to September 2016 this campaign. In addition, 25 RS were launched during this period, 6 of them during night-time in order to perform regular calibration of the Raman lidar water vapour channel.

## 2.2 IISTA-CEAMA station

One of the stations where this study has been carried out is the Andalusian Institute of Earth System Research (IISTA-CEAMA), an urban station of the University of Granada (UGR) located at Granada, Spain (37.16° N, 3.61° W, 680 m a.s.l.). This region is a complex terrain surrounded by mountains, mainly affected by Mediterranean continental climate conditions with hot summers and cool winters. Navas-Guzmán et al. (2014) analyzed one year of measurements of RH profiles at Granada showing that this location presents low values of RH (below 60%) in the 75% of the cases studied within 1.0 and 2.0 km a.s.l. This study also showed that mostly the cases with RH above 60% are found in spring and winter seasons. Regarding the remote aerosol sources, Granada is predominantly affected by aerosol particles coming from Europe and mineral dust particles from the African continent (Lyamani et al., 2006a, b; Guerrero-Rascado et al., 2008a, 2009; Córdoba-Jabonero et al., 2011; Titos et al., 2012; Navas-Guzmán et al., 2013; Valenzuela et al., 2014; Benavent-Oltra et al., 2017; Cazorla et al., 2017). Main local sources are road traffic, domestic- heating and biomass burning (mostly in wintertime) (Titos et al., 2017). Transported smoke principally from North America, North Africa and the Iberian Peninsula can also affect the study area (Alados-Arboledas et al., 2011; Navas-Guzmán et al., 2013; Preißler et al., 2013; Pereira et al., 2014; Ortiz-Amezcua et al., 2017). The probability of marine particles to reach the city is low taking into account that Granada is far away from the coast about 50 km in straight line, the marine particles would have to overpass some mountains in the path from the sea to the city and the air masses monitored over Granada are really dry. Also, Titos et al. (2014) showed that the contribution of marine aerosols to PM10 mass concentration was almost negligible (<3%) at IISTA-CEAMA station during the period 2006-2010. I addition, this work also refers to the identification of fine (PM1) and coarse (PM10) particulate matter in an urban environment of Granada.

The main instrument used in this study is the multiwavelength Raman lidar (RL) MULHACEN (Raymetrics S.A., Greece) located at IISTA-CEAMA station. MULHACEN is included in EARLINET (European Aerosol Lidar NETwork) (Pappalardo, et al., 2014), now operating in the framework of ACTRIS-2 (Aerosols, Clouds and Trace gases Research Infrastructure). It emits laser pulses at 355 and 532 nm (parallel and perpendicular polarization channels) and 1064 nm and it receives backscattered photons at 355, 532 and 1064 nm in analog and photon counting modes. Also, it collects Raman backscattered photons at 607 and 387 nm for molecular nitrogen ($N_2$) and at 408 nm for water vapor ($H_2O$) in photon counting mode, used for routine nighttime measurements. Such kind of configuration allows for deriving not only vertically-resolved particle information but also water vapour mixing ratio profiles. Vertical resolution for lidar backscattered signals is 7.5 m. Atmospheric information retrieved from lower regions is limited by the full overlap height, which is reached above 1.3 km a.s.l due to the system configuration (Guerrero-Rascado et al., 2010; Navas-Guzmán et al., 2011). A full description of this instrument can be found in Guerrero-Rascado et al. (2008a; 2009). Aerosol particle backscatter coefficient profiles $(\beta_{par}(z))$ are retrieved by the Klett-Fernald method (Fernald, 1984; Klett, 1981; 1985). The total uncertainty for $\beta_{par}$ retrieved with this method is usually within 20% (e.g. Franke et al., 2001; Preißler et al., 2011).

The ground-based MWR (RPG-HATPRO G2, Radiometer physics GmbH), which is included in MWRnet (Rose et al., 2005; Caumont et al., 2016), is used here for retrieving temperature profiles. The MWR is a passive remote sensor that performs automatic measurements of sky brightness temperature at two bands: the oxygen V band (51-58 GHz) and water vapor K band (22-31 GHz) associated to temperature and water vapor and liquid water, respectively. The MWR has a radiometric resolution between 0.3 and 0.4 RMS errors at 1.0 s integration time. The retrievals of temperature profiles from the measured brightness temperatures are performed using standard feed forward neural network (Rose et al., 2005). A detailed description of this system can be found in Granados-Muñoz et al. (2012) and Navas-Guzmán et al. (2014). Temperature data are provided at 39 height-bins, with variable vertical resolution. The first 25 bins are located below 2 km (mainly within the atmospheric boundary layer (ABL)) with a resolution ranging between 10 and 200 m, whereas the vertical resolution is much lower in the free troposphere (between 200 and 2000 m) with only 14 bins between 2 and 10 km. The accuracy and the precision of the temperature profiles of this radiometer were evaluated against RS by Bedoya et al. (2017). This study revealed differences between RS and the MWR temperature profiles lower than 0.5 K below 2.5 km and up to 1.7 K at higher altitude levels. Those results are within the accuracy of the temperature profile reported by the manufacturer, which is lower than 0.75 K RMSE (1.2-4.0 km range) and larger than 1.0 K RMS from 4 to 10 km.

Co-located RS are occasionally launched when Raman lidar measurements are taken. The RS data are obtained with a GRAW DFM-06/09 system (GRAW Radiosondes, Germany), which provides temperature (resolution 0.01°C, accuracy 0.2 °C), pressure (resolution 0.1 hPa, accuracy 0.5 hPa) and RH (resolution 1%, accuracy 2%) profiles with vertical resolution depending on the sonde ascension velocity, usually around 5 m/s. Data acquisition and processing are performed by the GRAWmet software and GS-E ground station from the same manufacturer.

A co-located Doppler lidar system (HALO photonics Stream Line) is also operating at IISTA-CEAMA station since May 2016. This system provides range-resolved measurements of attenuated backscatter based on the frequency shift associated to the movement of the particles and clouds in the atmosphere by means of heterodyne optical detection principle (Pearson et al., 2008). As this movement is linked with wind, the 3-D wind vector can be determined through the Doppler effect. Radial velocity measurements are taken every 2 s, and conical scans are performed every 10 min with 75º elevation angle and at 12 equidistant azimuth angles. The eye-safe laser transmitter vertically pointing to zenith operates at 1.5 μm, with low pulse energy (~100μJ) and high pulse repetition rate (~15 kHz) on a monostatic coaxial set up. See Päschke et al. (2015) for further information of the system configuration.

## 2.3 Sierra Nevada Station

At Sierra Nevada station (SNS, 37.09 N, 3.38° W, 2500 m a.s.l.), state-of-the-art in-situ instrumentation was operated to characterize aerosol properties. The inlet at SNS station is a whole air inlet located in the rooftop of a 3-story building. It is made up of stainless steel tube, with dimensions of 10 cm in diameter and 2.5 m in length. Inside the main tube there is a laminar flow of 100 litres per minute and there are several stainless-steel pipes that drive the sampling air to the different instruments. Each one of the stainless-steel pipes extracts the appropriate flow for each instrument. Different diameters of the pipes have been selected in order to optimize the efficiency of the system (Baron and Willeke, 2001). The instrumentation used in this study includes an Aerodyne Aerosol Chemical Speciation Monitor (ACSM, Aerodyne Research Inc.), an Aethalometer (AE33 model, Magee Scientific, Aerosol d.o.o.), an Aerodynamic Particle Sizer (APS, TSI 3321) spectrometer and a Scanning Mobility Particle Sizer (SMPS, TSI 3938) spectrometer; all of them connected to the main inlet. The ACSM was used to measure on-line submicron inorganic (nitrate, sulphate and ammonium) and organic aerosol (OA) concentrations. Equivalent black carbon, eBC, mass concentration was obtained from measurements of the Aethalometer AE33 at 880 nm. A mass absorption cross section of 7.77 $m^2g^{-1}$ was used to convert the absorption coefficients at 880 nm in eBC mass concentrations (Drinovec et al., 2015). Particle number size distributions were retrieved by a combination of the measurements performed with the SMPS in the diameter range 13-600 nm and the APS for the range 0.6-20µm.

## 3 Methodology

### 3.1 RH-profiles by synergy of RL and MWR data

As mentioned in section 2, some RL systems can provide simultaneous aerosol and water vapour profiles with high vertical and temporal resolution. The water vapour mixing ratio r(z) can be obtained from the ratio of Raman lidar signals of water vapour (408 nm) and nitrogen (387 nm) multiplied by a constant C that takes into account the fractional volume of nitrogen, the ratio between molecular masses, some range-independent constants and the Raman backscatter cross sections for nitrogen and water vapour molecules (Mattis et al., 2002). In the present study, the calibration constant C has been calculated using the simultaneous and collocated radiosondes launched at the EARLINET IISTA-CEAMA station during the analysed periods. C is obtained as the average value of the ratio between the uncalibrated RL r(z) profile and the r (z) profile from RS over the height-range from 1.5 to 3.5 km a.s.l. (Guerrero-Rascado et al., 2008b; Leblanc et al., 2012; Navas-Guzmán et al. 2014; Foth et al., 2015). C remains constant over periods when the lidar setup is not modified and the system presents good alignment, allowing us to retrieve r (z) profiles from the RL even when RS measurements are not available. If several RS launches are available during a certain period, C is obtained as the average between all calibrations performed over that particular period.

Temperature profiles from the MWR, which are continuously measured every 2 min, combined with 30 min- averaged r (z) profiles as proposed by Navas-Guzmán et al. (2014), are used to retrieve the RH profiles required for aerosol hygroscopic growth studies each 30 min. The following equation is used for retrieve the RH profiles

$$RH\ (z) = \frac{100 P\ (z)\ r\ (z)}{e_w\ (z)\ [621,97 + r\ (z)]} \tag{1}$$

where r (z) is obtained from the calibrated water vapour channel, P (z) (hPa) is the ground-scaled pressure profile and $e_w$ (z) is the water vapour pressure (hPa), calculated from the temperature profiles (List, 1951). Temperature profiles were scaled to lidar vertical resolution by linear interpolation.

## 3.2 Selection criteria for hygroscopic cases

A simultaneous increase in aerosol properties, such as particle backscatter ($\beta_{par}$) or extinction ($\alpha_{par}$) coefficients, and RH values over a certain atmospheric layer might be an indication of aerosol hygroscopic growth. Moreover, decreasing Ångström exponent (AE) and particle linear depolarization ratio (PLDR) are related to larger and more spherical particles, which also points to aerosol water uptake (Granados-Muñoz et al., 2015; Fernández et al., 2015; Haarig et al., 2017). However, additional

constraints need to be fulfilled when studying the aerosol hygroscopic growth in the atmosphere by remote sensing techniques due to the lack of control over the environmental conditions, as opposed to in-situ measurements. These constraints are used for guarantying that variations in the aerosol properties are due to water uptake and not to changes in the aerosol load or type.

A first constraint that needs to be satisfied is that the origin and pathways of the air masses arriving at different altitudes within the analyzed layer must be the same in order to avoid transport of different aerosol types from different source regions

(Veselovskii et al., 2009; Granados-Muñoz et al., 2015). The evaluation of the aerosol origin and transport is performed here through backward trajectories analysis using HYSPLIT model (Hybrid Single-Particle Lagrangian Integrated Trajectory) (Draxler and Rolph., 2003) with GDAS data as meteorological input with spatial resolution of 0.5°x0.5° available since 2010 with daily files every three hours on the native GFS hybrid sigma coordinate system. As a second constraint, atmospheric vertical homogeneity must be ensured. In order to evaluate the atmospheric vertical mixing, virtual potential temperature

$\left(\theta_v\ (z)\right)$ and r (z) profiles are analyzed. Low vertical variability of those variables suggests atmospheric vertical homogeneity in the layer of study (Veselovskii et al., 2009; Fernández et al., 2015; Granados-Muñoz et al., 2015; Lv et al., 2017). In addition, horizontal and vertical wind velocities and directions retrieved from the lidar Doppler system operated at Granada station were considered. Low horizontal wind velocity measured at different altitude levels is used as indicator of no particle advection into the layer analysed, taking into account that wind direction must be constant during long time periods (more than 3 hours). The

third moment of the frequency distribution of vertical wind velocities (skewness) has also been calculated in order to evaluate

convection of air masses within the column studied, having in mind that positive values of skewness represent upward wind velocity and negative values the opposite (O'Connor et al., 2010).

## 3.3 Relative humidity and aerosol properties

Once the requirements described in the section 3.2 are fulfilled, the cases of hygroscopic growth can be studied by means of the enhancement factor ($f_\xi(\lambda, RH)$), defined as follows:

$$f_\xi(\lambda, RH) = \frac{\xi(\lambda, RH)}{\xi(\lambda, RH_{ref})} \tag{2}$$

where $\xi(\lambda, RH)$ represents an aerosol optical/microphysical property evaluated at certain RH. The value of $RH_{ref}$ is taken from
10   each profile and corresponds to the lowest RH in the evaluated layer. In this study, the optical property used is $\beta_{par}$ at 355 and 532 and, thus, the backscatter enhancement factor is denoted as $f_\beta(\lambda, RH)$. Estimations of $f_\beta(\lambda, RH)$ uncertainty are very scarce because of their high complexity. Some studies (e. g. Adam et al., 2012; Zieger et al., 2013) provided estimations based on sensitivity analysis using Mie model calculations, reporting errors around 20% on $f_\sigma(\lambda, RH)$, where $\sigma$ is the scattering coefficient. Titos et al. (2016) reported uncertainty estimations based on Monte-Carlo techniques, concluding that the more
15   hygroscopic the aerosol, the higher is the uncertainty in $f_\sigma(\lambda, RH)$ especially at high RH ($RH > 80\%$). For moderate-hygroscopic aerosol, it was established a lower limit for the uncertainty in $f_\sigma(\lambda, RH)$ of around 30-40% using nephelometry techniques.

In aerosol hygroscopic growth studies, humidograms are usually parameterized by using fitting equations (e.g. Titos et al.,
20   2016) of varying complexity. One of the most common used parameterizations is the one-parameter equation introduced by Hänel et al., (1976):

$$f_\beta^\lambda(RH) = \left(\frac{(1-RH)}{(1-RH_{ref})}\right)^{-\gamma(\lambda)} \tag{3}$$

where $\gamma$ is a parameter related to the aerosol hygroscopicity. This parameter depends on the aerosol type and wavelength.

## 3.4. Mie model to calculate enhancement factor at SNS

In order to validate the results obtained with the remote sensors for $f_\beta^\lambda(RH)$ and $\gamma(\lambda)$, theoretical calculations based on Mie theory (Mie, 1908) have been performed using data from SNS in-situ instrumentation as input for Mie model. The particle backscatter coefficients at dry and humid conditions have been calculated with a model based on Mie theory where the core

Mie routine is based on the code of Bohren and Huffmann (2004). The particles are assumed to be spherical and homogenously internally mixed. For this analysis, the particle number size distribution and the complex refractive index m of the measured aerosol is needed as input. We calculated the aerosol complex refractive index using the chemical composition measured with the ACSM combined with the black carbon (eBC) mass concentration from the aethalometer. Then, the refractive index was determined by a volume fraction averaging:

$$m(\lambda) = \rho \sum \frac{F_i}{\rho_i} m_i(\lambda) \tag{4}$$

where $\rho$ is the total density of the aerosol, $F_i$ is the mass fraction, $\rho_i$ is the density and $m_i(\lambda)$ is the wavelength-dependent complex refractive index of the compound i. The values of $\rho_i$ and $m_i(\lambda)$ are taken from the literature and are listed on Table 1.

Hygroscopic growth was also accounted for by considering the aerosol chemical composition measured with the ACSM and the eBC mass concentrations measured with the Aethalometer. For this, we used the individual growth factor g(RH) as reported in Table1. These g (RH) were extrapolated to different RH using Eq. (3) from Gysel et al. (2009), which uses the κ-model introduced by Petters and Kreidenweis (2007). A mean g (RH) is then calculated with the Zdanovskii-Stokes-Robinson relationship (Stokes and Robinson, 1966) from the g (RH) of the individual components of the aerosol and their respective volume fractions. For the wet refractive index, a volume weighting between the refractive indices of the dry aerosol and water was used (Hale and Querry, 1973).

## 4. Results and discussion

### 4.1. Combination of RL and MWR method for retrieving RH profiles

The synergetic method proposed by Navas-Guzmán et al. (2014) for retrieving RH profiles is used here for the first time to study aerosol hygroscopic growth. In this section, two particular cases (case I on 22nd July 2011 at 20:00-20:30 UTC and case II on 22ndJuly 2013 at 20:30-21:00 UTC) are analysed with this new methodology. These two cases were already presented in Granados-Muñoz et al. (2015) using the classical approach that combines RH profiles obtained from RS and the lidar aerosol properties. Results obtained here are compared with those in Granados-Muñoz et al. (2015) in order to evaluate the synergetic method proposed here.

Figure 2 shows, from left to right, the RH profiles obtained from both the RS (black line) and the synergy RL+MWR (red line), the bias between both profiles (RH $_{RS}$ − RH $_{RL + MWR}$), $\beta$ $_{532nm}$ profiles retrieved from the lidar system and $f_{\beta}$ (RH). Values of $f_{\beta}$ (RH) obtained in Granados-Muñoz et al. (2015) using RS measurements are shown in black, whereas the red line

corresponds to the synergy RL + MWR presented here. The top row corresponds to case I on 22$^{nd}$ July 2011 and the bottom row to case II on 22$^{nd}$ July 2013. Horizontal dashed lines mark the region of interest analysed for each case, ranging from 1.3 to 2.3 km a.s.l. for case I and 1.3 to 2.7 km a.s.l. for case II.

RH profiles (Fig. 2 a and e, red line) calculated by the combination between RL calibrated r (z) profile and MWR temperature profiles were obtained following the methodology presented in section 3.1 by using Eq. 3 (Navas-Guzmán et al., 2014). Good agreement is observed, with biases (Fig. 2 b and f) lower than 10% within the analysed region. The differences obtained in the RH profiles might be associated to the discrepancies between the temperature profiles from MWR and RS, due to the lower vertical resolution of the MWR. Additionally, discrepancies are also expected because of the radiosonde drift and the different

temporal sampling (the lidar data correspond to a 30-min average, whereas the RS provides instantaneous values that build the profile in the region of interest in less than 5 minutes).

The discrepancies between the two RH profiles are especially relevant in the lower part of the analyzed data since differences of RH in this region lead to variations in RH $_{ref}$. For case I, RH $_{ref}$ = 60% for RS and RH $_{ref}$ = 68% for the RL + MWR

combination, whereas for case II, RH $_{ref}$ = 40% for RS and RH $_{ref}$ = 50% for RL + MWR methodology. Additionally, the RH discrepancies in the upper region of the profiles (from 2.1 to 2.3 km a.s.l. for case I and from 2.6 to 2.7 km a.s.l. for case II), which can reach up to 5%, are also relevant since they are associated to the maximum values of RH and may modify the data tendency on Hänel's parameterization, leading to variations in $\gamma (\lambda)$ depending on the methodology used for the retrieval of RH. Despite these discrepancies, the differences between $\gamma (\lambda)$ parameters obtained from both methodologies are low (Table

2). On case I, $\gamma (\lambda) = 0.59 \pm 0.05$ obtained from RL + MWR is larger than that obtained from RS ($\gamma = 0.56 \pm 0.01$). On case II the RS $\gamma = 0.99 \pm 0.01$ is larger than $\gamma = 0.95 \pm 0.02$ obtained from RL+MWR, having in mind that uncertainties reported on $\gamma$ are obtained by the polynomial fitting but they do not include the propagation error result. The relative differences on both cases are below 5%, which is relatively good compared to the expected uncertainties reported in Titos et al., (2016) and considering the differences between the two methodologies.

The obtained values of f$_\beta$ (85%) using both methodologies are presented in Table 2. For case I, f$_\beta$ (85%) = 1.50 for RS and f$_\beta$ (85%) = 1.46 for RL+MWR, with a relative difference below 3%. For case II, f$_\beta$ (85%) = 2.6 for RS and f$_\beta$ (85%) = 2.3 for RL + MWR showing a relative difference of 11%. Even though the relative difference is larger for case II, for both cases the discrepancies lie within the uncertainty associated to the calculation of f$_\beta$ (85%) which is around 20% according to

Titos et al. (2016). Thus, the RL + MWR methodology presented by Navas-Guzmán et al. (2014) to obtain RH profiles in a continuous time base is a promising technique for hygroscopic growth studies. This methodology will allow for expanding the RH profiles database and it opens new opportunities for the detection of hygroscopicity cases during night time periods.

## 4.2. Hygroscopic study during SLOPE I

### 4.2.1 Conditions for hygroscopic growth

5    Aerosol hygroscopic growth was observed during SLOPE I campaign in 2016 combining the remote sensing instruments and the RS. Fig. 3 shows the time series of range corrected signal (RCS) at 532 nm derived by the EARLINET lidar system at the IISTA-CEAMA station. The presence of clouds is observed in the late afternoon (~3.0 km a.s.l.) before 19:00 UTC, with clouds vanishing after that during the remaining measurement period. The red lines in Fig. 3 mark the 30 min set of profiles (from 20:30 to 21:00 UTC) where an intensification of the RCS is observed, an indication of potential aerosol hygroscopic 10   growth.

Fig. 4 shows profiles of $r$ (z), $\theta_v$, RH, $\beta_{par}$ at 355 and 532 nm, backscatter-related Ångström exponent between 355 and 532 nm (AE $_{355-532}$) and PLDR $_{532}$ (particle linear depolarization ratio at 532 nm) on $16^{th}$ June 2016 between 20:30 and 21:00 UTC. As we mentioned in section 3.2, for aerosol hygroscopic analysis it must be ensured that ranges where RH increases 15   correspond to increases in $\beta_{par}$, which is well seen along the layer between 1.5 and 2.4 km a.s.l. (see Fig. 3). The RH profile was calculated by using the method RL + MWR. In this case, the calibration constant for the RL $r$ (z) was calculated using the six RS launched at night-time during SLOPE I campaign. The calibration constant was $110 \pm 2$ g/kg, calculated as the mean value for the months of May-June-July, when no changes were performed on the RL system.

20   In order to fulfil the constraints presented in sections 3.2 and 3.3 for hygroscopic growth studies, RH and $\beta_{par}$ must increase within the layer evaluated but also the atmospheric stability must be ensured through the evaluation of thermodynamic variables such as $\theta_v$ and $r$ (z). Here, $r$ (z) shows relatively low variation within the region of interest (1.5 to 2.4 km a.s.l.), decreasing monotonically with altitude at a rate of $-1.9 \frac{g}{kg \cdot km}$ (Fig. 4a) and $\theta_v$ shows a monotonic increase

rate $\left(\frac{\partial \theta_v}{\partial z} = 0.03 \frac{°C}{km}\right)$ within the same region.

AE $_{355-532}$ and PLDR $_{532}$ were also retrieved in order to describe the size and shape of the aerosol particles. For this case, we observe a decrease in both parameters in the region of interest. A decrease in AE $_{355-532 \, nm}$ (~0.4 km$^{-1}$) means an increase in the predominance of larger particles and a decrease of the PLDR $_{532 \, nm}$ (~0.13 km$^{-1}$) is related to particles becoming more spherical. This correlation between AE $_{355-532}$ and PLDR has been observed in previous studies associated to hygroscopic 30   growth (Granados-Muñoz et al., 2015; Haarig et al., 2017).

In order to determine the origin of the aerosol particles over the analysed layer, we present a horizontal wind speed and direction and vertical wind analysis from Doppler lidar data. The 10min-resolved horizontal wind direction time series (Fig. 5b) indicate

that from 18:00 to 21:00 UTC the wind over IISTA-CEAMA station mainly came from the North-West, within the region of interest (1.5 to 2.4 km a.s.l.) with relative low horizontal wind velocity (up to 6 m/s) (Fig. 5a), which means that aerosol particles were being transported from the same direction, likely coming from the same source, at relative low horizontal velocity.

A turbulence analysis was also performed to reinforce the fact that vertical fluxes within the aerosol column are associated to increases of RCS observed on Fig. 3. Furthermore, the aerosol RCS increases in a region where RH increases as we see on Fig. 4, leading us to associate the increase in RCS with water uptake by aerosols inside the column. The vertical wind velocity can be statistically studied to obtain the higher moments of the velocity distribution (O'Connor et al., 2010, Moreira et al., 2018a). This statistical analysis is deeply developed for turbulence studies. Here the third moment of the frequency distribution (skewness) (Fig. 5c) represents the direction of the convection (positive skewness is associated to predominance of upward wind velocity whereas negative skewness means predominance of downward wind) in the region of interest. Supporting this analysis, the black stars represent the calculation of the atmospheric boundary layer height (ABLH, Fig. 5 c) obtained from the MWR data by using the combination of parcel and gradient methods in convective and stable atmospheric conditions (Holzworth, 1964; Moreira et al., 2018b). In this case, close to 21:00 UTC (Fig. 5 c), the particles tend to ascend into the column, as indicated by positive values reached in the skewness linked with highly convective movement. The ABLH reach its maximum at 15:00 UTC (2.5 km a.s.l.) but after 16:00 UTC the weakening of convection tends to decrease the ABLH, keeping the ABLH around 2 km a.s.l. until 21:00 UTC. All this wind information might be interpreted as transported particles coming from the same direction at relative low horizontal velocities, suggesting that aerosol source is not changing and new aerosol particles are not advected into the studied layer. The turbulence analysis allows us to support that vertical wind movement within the layer of interest drives to well mixing processes during the analysed time interval.

The 6-day backward trajectories were calculated at three different heights (900, 1500 and 1900 m a.g.l), which were selected within the region of interest in order to guarantee the height-independency of the air masses pathway. The three air masses came from North America, crossing the Atlantic Ocean, reaching the continental platform through Portugal and then advected to Granada reaching the station at 21:00 UTC (not shown here). This information supports the horizontal wind analysis performed before.

### 4.2.2 $f_\beta^\lambda$ (RH) measured and retrieved by combining in-situ data and Mie theory

The humidogram presented in Fig. 6 shows the measured $f_\beta^\lambda$ (RH)at 355 and 532 nm as a function of RH between 1.5 and 2.4 km a.s.l. The calculated $f_\beta^\lambda$(RH) obtained with Mie theory and the measured chemical composition and size distribution at SNS station (2.5 km a.s.l.) is also shown in Fig.6. The humidogram exhibits a monotonic positive slope at both wavelengths, for

RH between 78 and 98%. The RH $_{ref}$ = 78 % was selected as the lowest RH value into the evaluated column, and this same RH was used as reference for the Mie calculation in order to make both calculations comparable. With our model based on Mie theory we calculated the $f_\beta^\lambda$ (RH), which depends on the aerosol chemical composition and size distribution and on the ambient RH. Using the values reported in Table 1, the measured chemical composition and particle number size distribution as inputs for the model, the $f_\beta^\lambda$ (RH) at SNS was obtained.

At SNS station, according to the mean size distribution calculated with SMPS during the hygroscopic growth case two main peaks were detected at 35 and 115 nm (from 20:00 to 21:00 UTC), therefore, it can be observed that most of the aerosol is in the fine mode (< 1 µm), but the lidar's wavelengths used are more efficient at those wavelengths. The sub-micron mass concentration measured with ACSM indicates high concentration of organic particles during daytime (from 12:00 to 17:00 UTC), with values around 7 µg/m$^3$ at 15:00 UTC. OA concentrations decreased slowly to values around 3.0 µg/m$^3$ at 00:00 UTC. In particular, during the hygroscopic growth case under study (from 20:00 to 21:00 UTC) the aerosol composition was mainly made up of organic particles (62%) followed by sulphate (24%), nitrate (10%), ammonia (2%) and black carbon (2%). Thus, the predominant aerosol studied during the event is a composition of smoke and urban polluted aerosol. This assumption about the aerosol type is supported by the relatively high sulphate concentration observed at SNS and the results discussed in section 4.2.1 (lidar properties, backward trajectories analyses). The backscatter-related AE values (close to 1.5) indicates a predominance of fine particles which is connected with the presence of smoke over that comes from North America as suggested by the backward trajectories analyses. This chemical composition with high predominance of organic particles is consistent with the $\gamma$ values obtained with the RL + MWR method. Fernández et al. (2015) reported a similar $\gamma^{532}$ value of 0.59 in Cabauw (Netherlands) associated with high concentration of organic particles while they observed a significantly larger $\gamma^{532}$ of 0.88 associated with marine particles. Lower values are reported by Lv et al. (2017) in one of their case studies ($\gamma^{532}$ = 0.24 and $\gamma^{355}$ = 0.12) in Xinzhou (China) associated with the presence of dust particles. Although the behaviour of the backscatter coefficient at enhanced RH is expected to differ from that of the scattering coefficient, a qualitative comparison can be performed due to the scarcity of backscatter-related $\gamma$ values in the literature. For example, using in-situ techniques, Zieger et al. (2015) reported a low scattering enhancement of boreal aerosol in Hyytiälä (Finland) ($\gamma^{525}$ = 0.25) related to the high contribution of organic aerosols at this site that contribute to decrease the hygroscopic enhancement. At Cape Cod (USA), Titos et al. (2014) reported significantly lower $\gamma$ values for polluted aerosols ($\gamma^{550}$ = 0.4 ± 0.1) compared with marine aerosols ($\gamma^{550}$ = 0.7 ± 0.1).

Calculated and measured values of $f_\beta^\lambda$ (RH) are compared in Table 3 and Figure 6. In general, there is a good agreement between measured and calculated hygroscopicity parameters. For both wavelengths, slightly higher values are predicted by the model compared with the measurements, especially at RH>90% where the differences are higher than at RH < 90%. The values retrieved with RL are $f_\beta^{355}$ (85%) = 1.07 ± 0.03 and $f_\beta^{532}$ (85%) = 1.20 ± 0.03 and with Mie theory are $f_\beta^{355}$ (85%) =

1.10 $\pm$ 0.01 and $f_\beta^{532}$ (85%) = 1.15 $\pm$ 0.01. The good agreement found in this analysis has relative differences lower than 4 % taking calculated data as reference. The $\gamma$ shows good agreement between the measured ($\gamma^{532}$ = 0.48 $\pm$ 0.01 and $\gamma^{355}$ = 0.40 $\pm$ 0.01) respect to Mie calculated ($\gamma^{532}$ = 0.53 $\pm$ 0.02 and $\gamma^{355}$ = 0.45 $\pm$ 0.02), with relative differences between them of 9 % at 532 nm and 11 % at 355 nm, taking the calculated values as reference. The good agreement between the

measured and theoretical backscatter enhancement factor evidences the robustness of the proposed method for hygroscopic studies in a systematic manner.

The principal sources of error in the comparison between calculated and measured data are associated with the method for the retrieval of RH profiles itself, as well as the errors associated with theoretical Mie calculation mainly by the assumption of

g (RH) based on the chemical composition. Finally, the horizontal distance between stations (SNS and IISTA-CEAMA) could also be a reason of the discrepancies found here. The uncertainties involved here are hard to calculate because the contributions of the particle backscatter uncertainties and experimental uncertainties associated to determination of the backscatter enhancement factor, thus further studies should center the efforts to this research field.

In addition, the multiwavelength results lead us to see a clear spectral dependence on $\gamma$ ($\lambda$). The efficiency due to changes in $f_\beta^\lambda$ (RH) associated to $\beta_{par}$ and to RH is stronger at 532 nm than at 355 nm, finding that $f_\beta^{532}$ (85%) = 1.20 > $f_\beta^{355}$ (85%) = 1.07, and also it is seen on gamma parameter $\gamma^{532}$ = 0.48 $\pm$ 0.01 > $\gamma^{355}$ = 0.40 $\pm$ 0.01 with correlations of 0.84 and 0.65, respectively. The wavelength dependency has also been reported in Kotchenruther et al. (1999) for in-situ measurements at 450, 550 and 700 nm, obtaining increasing enhancement factors with wavelength and in Zieger et al. (2013), where the same

behaviour is observed for marine aerosols. As it is reported in Haarig et al. (2017) the enhancement factor dependency with wavelength suggests that larger wavelengths have an enhancement factor larger than short ones which in fact was also evidenced on this work.

**5. Conclusions**

The methodology proposed for calculating RH profiles by combining calibrated r (z) from EARLINET RL and temperature from MWR profiles has been used for hygroscopic growth studies. With this method, a way to retrieve RHprofiles without the necessity of co-located RS is presented at IISTA-CEAMA station. In order to validate this methodology, hygroscopic growth

cases which use RS data were selected. The relative differences on the $f_\beta^\lambda$ (RH) obtained using the RH profiles from the RS and from the combined RL and MWR were calculated, finding relative differences below 11 % on $f_\beta$ (85%). The relative differences on $\gamma$ were below 5 %, supporting the fact that this methodology is valid for aerosol hygroscopicity studies.

Aerosol hygroscopic growth observed during SLOPE I field campaign (16th June 2016, 20:30 to 21:00 UTC) was studied by means of particle backscatter coefficient retrieved from the EARLINET multiwavelength RL, backscatter-related-Ångström exponent ($AE_{355-532}$) and particle linear depolarization ratio ($PLDR_{532}$) as optical properties and the combined RL + MWR RH profiles. Stability analysis confirmed good mixing conditions in the atmospheric layer studied. In addition, Doppler wind lidar data analysis allowed us to evaluate the vertical profiles of horizontal wind velocity and direction. Thus, we concluded that particles came mainly from the North-West region of Granada at low velocities. Furthermore, the skewness analysis let us infer that particles presented an upward movement during the 30 min evaluated period within the column of interest. These results were confirmed by ABLH calculations from MWR data. From the experimental data from RL, values of $f_\beta^{355}$ (85%) = $1.07 \pm 0.03$ and $f_\beta^{532}$ (85%) = $1.20 \pm 0.03$ at $RH_{ref}$ = 78 % were obtained within the evaluated column and also $\gamma^{532}$ = $0.47 \pm 0.01$ ($R^2$=0.84) and $\gamma^{355}$ = $0.40 \pm 0.01$ ($R^2$ = 0.65), which were in agreement with the literature.

For the study during SLOPE I the results were validated against Mie simulations with experimental data from SNS data obtaining a good agreement between the values retrieved with RL ($f_\beta^{355}$ (85%) = 1.07 and $f_\beta^{532}$ (85%) = 1.20) and Mie theory ($f_\beta^{355}$ (85%) = 1.10 and $f_\beta^{532}$ (85%) = 1.15) reaching relative differences lower than 4% taking calculated data as reference. We also found good agreement on γ parameters measured ($\gamma^{532}$ = $0.48 \pm 0.01$ and $\gamma^{355}$ = $0.40 \pm 0.01$) respect to calculated ($\gamma^{532}$ = $0.53 \pm 0.02$ and $\gamma^{355}$ = $0.45 \pm 0.02$), with relative differences between measured and calculated up to 9 % at 532 nm and 11 % at 355 nm, taking calculated data as reference. These results show that under favorable atmospheric conditions (vertical homogeneity, same origin of the aerosols within the analyzed layer, low horizontal velocity) and in absence of advected air masses into the evaluated column, the hygroscopic behavior of the particles evaluated by remote sensing at IISTA-CEAMA station is in accordance with that evaluated for those particles transported to SNS.

The results obtained here show the potentiality of combining a calibrated water vapor RL channel with MWR, making it possible to have RH profiles with high temporal/spatial resolution without co-located RS to analyze hygroscopic growth. These results will allow us to expand the database of hygroscopic growth cases studied with remote sensing techniques. With the proposed procedure the aerosol properties and the RH are obtained within the same atmospheric column, as opposed to the cases when the thermodynamic profiles are retrieved from RS.

**Acknowledgements**

This work was supported by the Andalusia Regional Government through project P12-RNM-2409, by the Spanish Ministry of Economy and Competitiveness through project CGL2013-45410-R, CGL2016-81092-R and through grant FPI (BES-2014-068893), the grant for PhD studies in Colombia, COLCIENCIAS (Doctorado Nacional - 647) associated to the Physics Sciences program at Universidad Nacional de Colombia, Sede Medellín and Asociación Universitaria Iberoamericana de

Postgrado (AUIP), Consejeria de Innovación, ciencia y Empleo de la Junta de Andalucía, also by "Juan de la Cierva-Formación" program (FJCI-2014-22052 and FJCI-2014-20819) and the Marie Skłodowska-CurieIndividual Fellowships (IF) ACE_GFAT (grant agreement No 659398). Also, it is supported by the Spanish Ministry of Economy and Competitiveness through the project CGL2013-45410- R, the University of Granada through the contract "Plan Propio P9, Convocatoria 2013".

This study was supported by the Swiss National Science Foundation Ambizione project "Study of aerosol hygroscopic effect on optical and microphysical properties by means of remote sensing techniques" (PZ00P2_168114). The financial support for EARLINET in the ACTRIS Research Infrastructure Project by the European Union's Horizon 2020 research and innovation program through project ACTRIS-2 (grant agreement No 654109). We thank the AERO group from ESRL-GMD at NOAA for providing the CPD software used for routine measurements at SNS and UGR stations, and for their technical support. The

authors thankfully acknowledge the FEDER program for the instrumentation used in this work.

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

**Table captions**

**Table 1.** Aerosol properties of selected compounds used for the model predictions, the refractive index ($m$) at 355 and 532 nm, density ($\rho$)and growth factor.

**Table 2**. Results obtained for each case analysed by means of new methodology combining RL+MWR and the classical approach using RS data.

**Table 3**. Results obtained for hygroscopic case on 16[th] June 2016, evaluated with RL (IISTA-CEAMA station) and in-situ (SNS) stations.

**Figure captions**

**Figure 1**. Vertical profile of SNS. The yellow star refers to IISTA-GRANADA station and green star refers to SNS.

**Figure 2**. (a, e) Profiles of $RH$ retrieved from $RS$ (black line) and by the synergy RL+MWR (red line), (b, f), $RH$ bias profiles (cyan line), (c, g) $\beta_{par}$ retrieved by using Klett-Fernald algorithm and LR=65 Sr (green line), (d, f) $f_\beta(RH)$ calculated for RS (black dots) and by the synergy RL+MWR (red dots) and the corresponding Hänel parameterizations (solid lines), where red line refers to RL+MWR method (Case I: $\gamma = 0.59 \pm 0.05$, Case II: $\gamma = 0.95 \pm 0.02$) and black line refers to RS method (Case I: $\gamma = 0.56 \pm 0.01$, Case II: $\gamma = 0.99 \pm 0.01$) .The top row corresponds to Case I (22[nd] July 2011, 20:30-21:00 UTC) and the bottom row to Case II (22[nd] July 2013, 20:00-20:30 UTC). Horizontal dashed lines indicate the altitude range analysed for each case (1.3 to 2.3 km for Case I and 1.3 to 2.7 km for Case II). All these profiles were measured at the EARLINET IISTA-CEAMA station.

**Figure 3.** EARLINET IISTA-CEAMA lidar $RCS$ time series at 532 nm, 16[th] June 2016 (17:00 to 00:00 UTC). The sunset estimated for this day was at 21:30 hour of local time.

**Figure 4.** (a) Water vapour mixing ratio; (b) virtual potential temperature; (c) relative humidity obtained from synergy RL+MWR; (d) particle backscatter coefficient at 355, 532; (e) backscatter-related Ångström exponent (355-532 nm) and (f) particle linear depolarization ratio. All profiles correspond to a 30 min-average from 20:30 to 21:00 UTC on 16[th] June 2016 at the EARLINET IISTA-CEAMA station.

**Figure 5.** Time series of (a) horizontal wind velocity, (b) horizontal wind direction and (c) skewness retrieved from Doppler turbulence calculations for 16th June 2016 at 20:30 to 21:00 UTC. The ABLH retrieved from MWR is presented in black stars.

**Figure 6.** Humidograms calculated (a) at 532 nm and (b) at 355 nm, within 1.5 to 2.4 km a.s.l. aerosol layer from the RL+MWR measurements and calculated using Mie theory and measured chemical composition and size distribution at 2.5 km a.s.l. A $RH_{ref} = 78$ % was used for both methods.

**Table 1:**

|  | $H_2O$ | OA | $NH_4NO_3$ | $(NH_4)_2SO_4$ | BC |
|---|---|---|---|---|---|
| $m$ **(355 nm)** | 1.343[a] | 1.458[g] | 1.562[c,f] | 1.56[e] | 1.75 +0.465i[d] |
| $m$ **(532 nm)** | 1.333[a] | 1.411[g] | 1.556[c] | 1.530[c] | 1.75 + 0.44i[d] |
| $\rho$ | 1 | 1.4[h] | 1.72[i] | 1.77[i] | 1.7[b] |
| $g$ **(RH=90%)** | -- | 1.05[j] | 1.74[k] | 1.66[k] | 1[l] |

(a) Hale and Querry (1973); (b) Nessler et al. (2005); (c) Fierz-Schmidhauser et al. (2010); (d) Hess et al. (1998)

(e) Ma and Thompson (2012); (f) Linear interpolation to 355 nm (Kou et al., 1993); (g) Nakayama et al. (2010)

(h) Alfarra et al. (2006); (i) Lide (2008); (j) Riipinen et al. (2015) for Dp = 100 nm; (k) Gysel et al. (2007) for Dp = 60 nm;

(l) BC was assumed to be insoluble (e.g. Hung et al., 2015).

**Table2:**

|  | Case I | Case II |
|---|---|---|
| **RS: $RH_{ref}$ [%]** | 60 | 40 |
| **RL+MWR: $RH_{ref}$ [%]** | 68 | 50 |
| **RS: $f_\beta$(85%)** | 1.50 | 2.60 |
| **RL+MWR: $f_\beta$(85%)** | 1.46 | 2.30 |
| $\gamma_{RS}$ | 0.56 ±0.01 | 0.99±0.01 |
| $\gamma_{RL+MWR}$ | 0.59 ±0.05 | 0.95±0.02 |

**Table 3:**

|  | **Measured** | **Calculated** |
|---|---|---|
| $RH_{ref}$ [%] | 78 | 78 |
| $f_\beta^{532}(85\%)$ | 1.20 | 1.15 |
| $f_\beta^{355}(85\%)$ | 1.07 | 1.10 |
| $\gamma^{532}$ | 0.48 ±0.01 ($R^2$=0.84) | 0.53±0.02 ($R^2$=0.94) |
| $\gamma^{355}$ | 0.40 ±0.01($R^2$=0.65) | 0.45±0.02 ($R^2$=0.93) |

**Figure 1:**

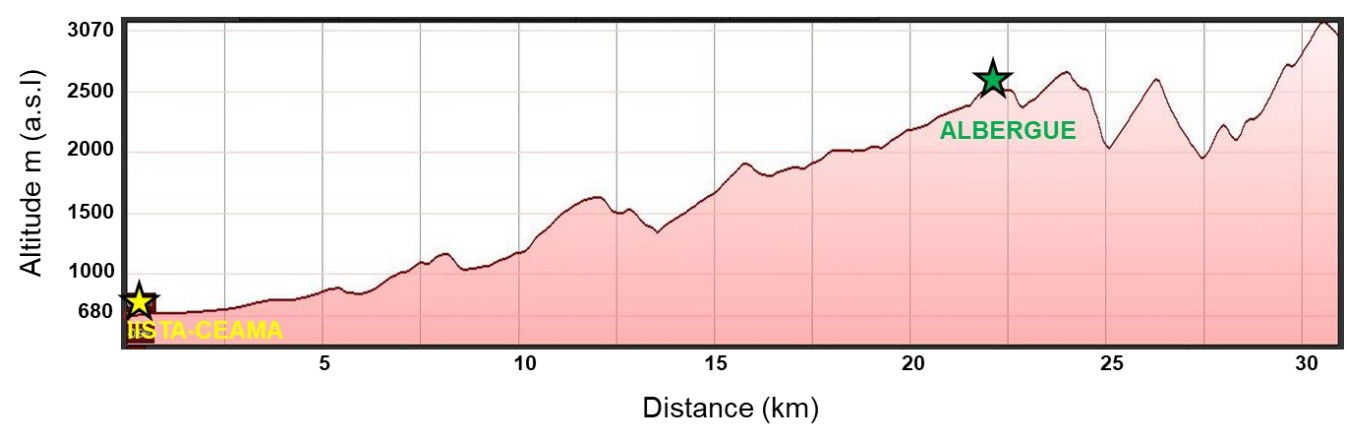

**Figure 2:**

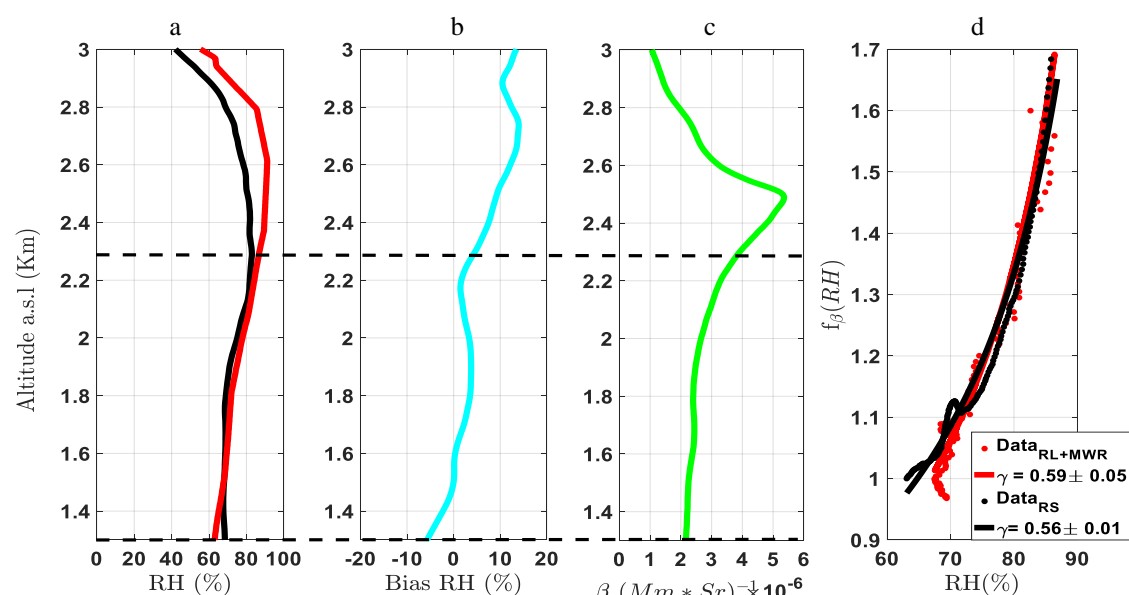

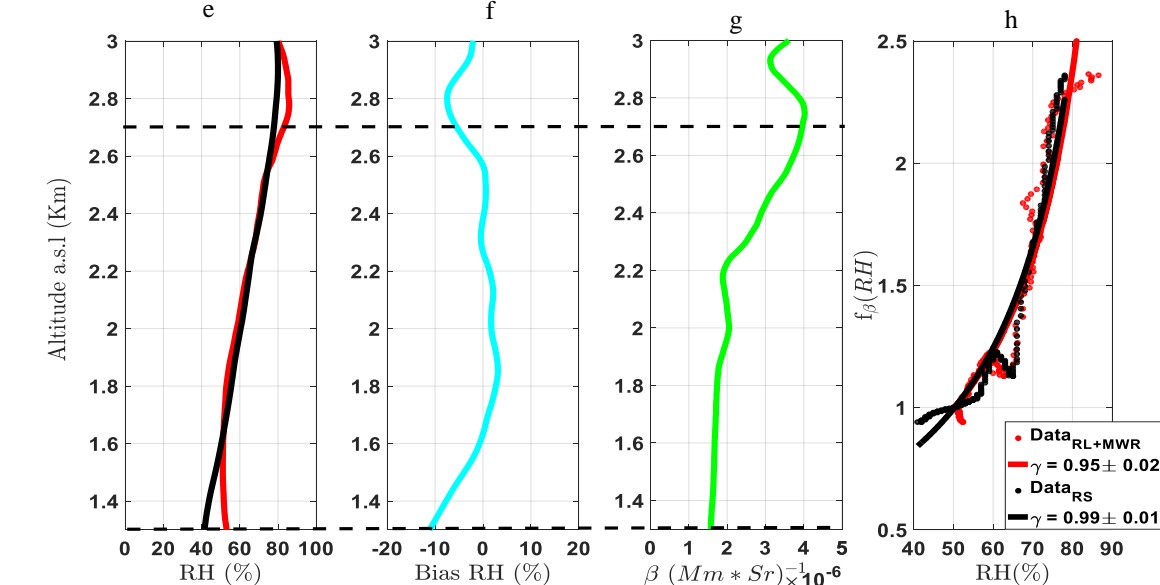

**Figure 3:**

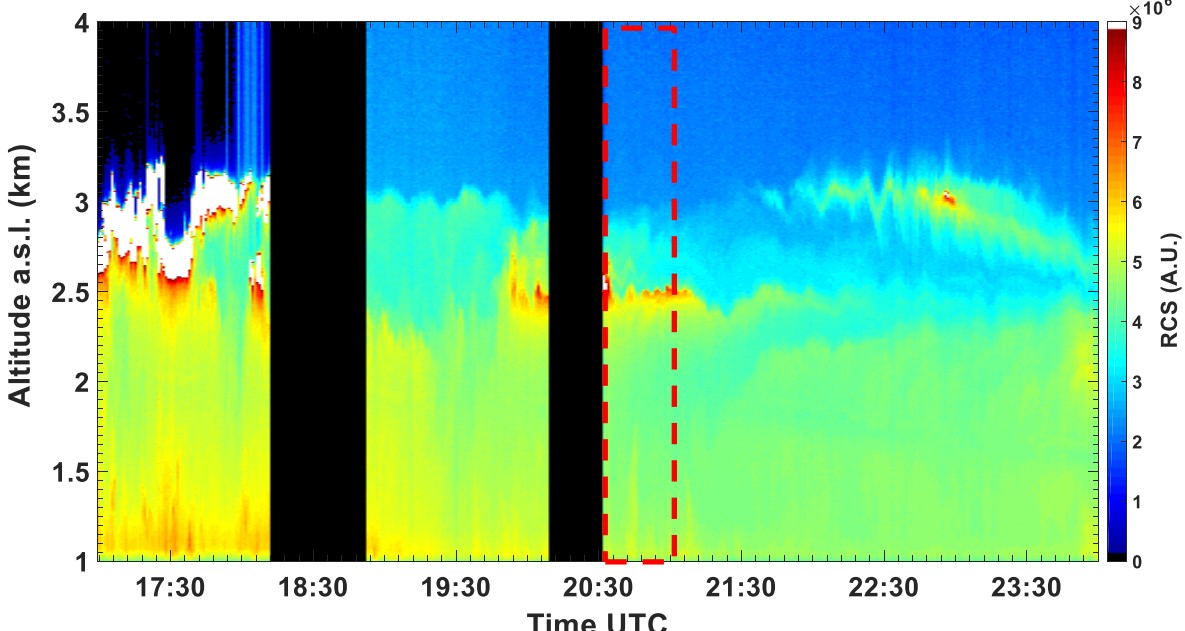

**Figure 4:**

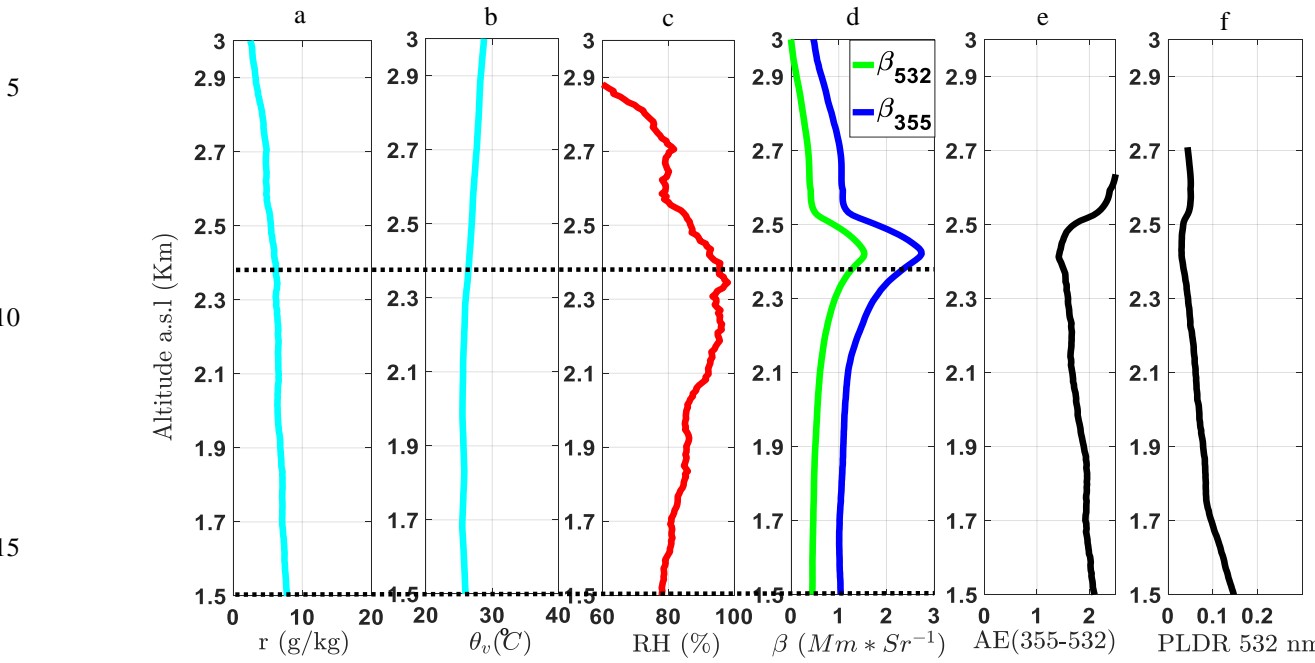

**Figure 5:**

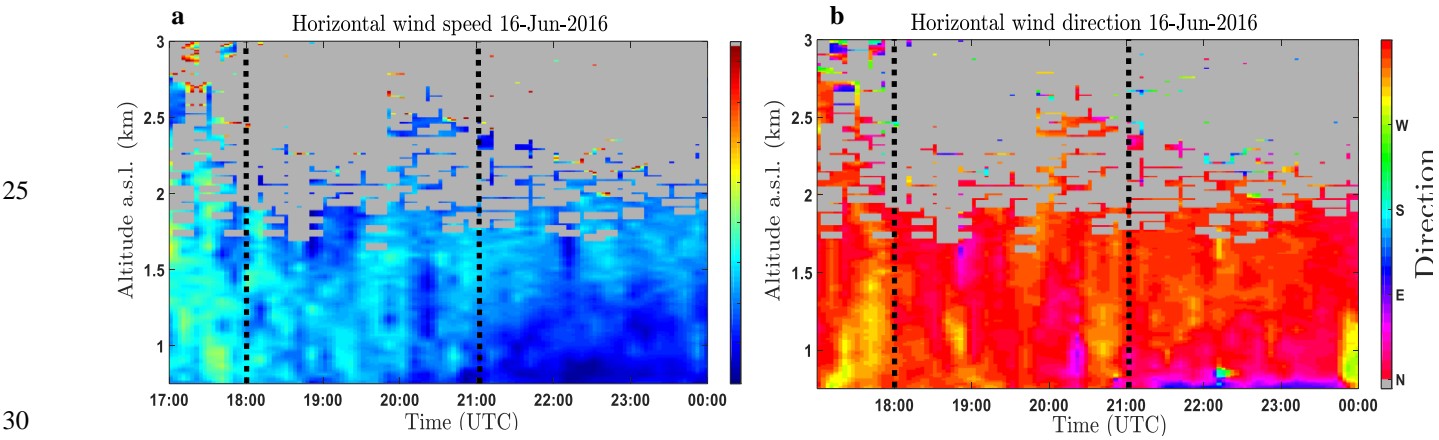

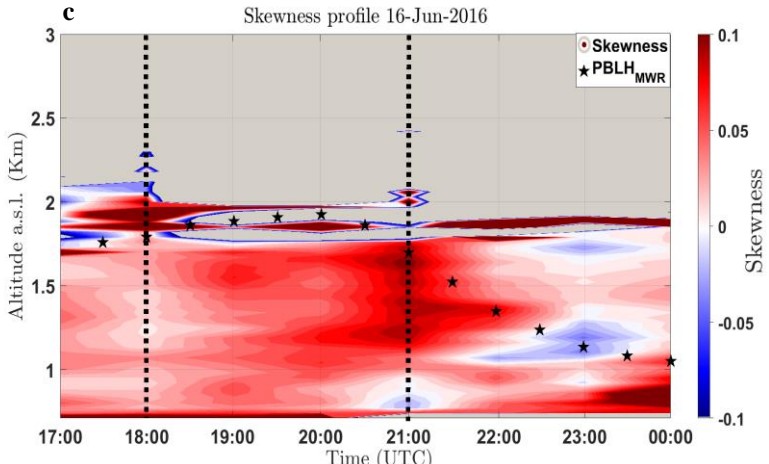

**Figure 6:**

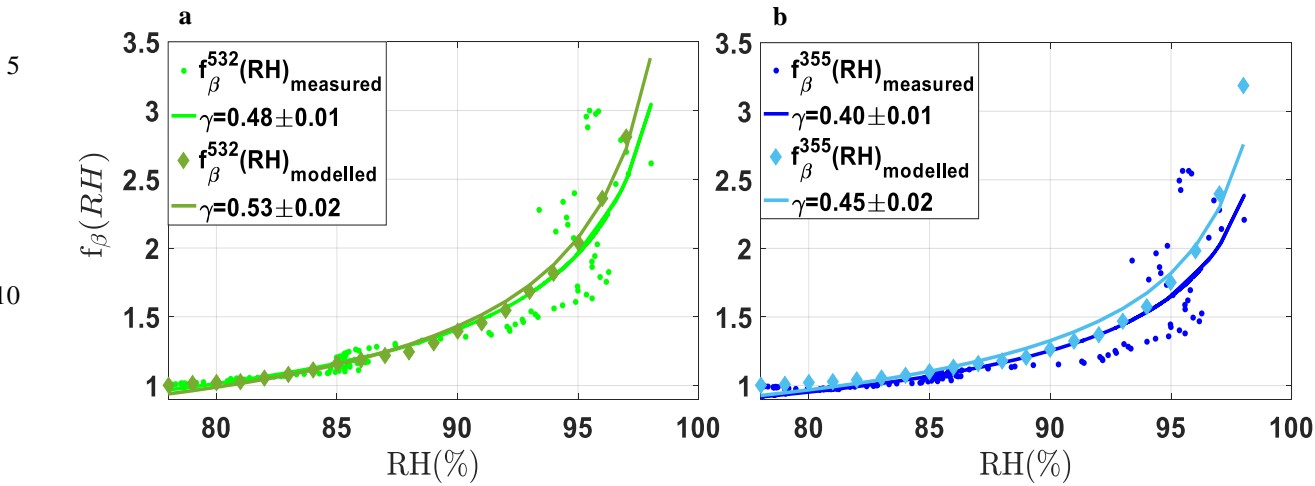