# Peer review of "Hygroscopic growth study in the framework of EARLINET during the SLOPE I campaign: synergy of remote sensing and in-situ instrumentation"

_Atmospheric Chemistry and Physics, 2017_

## Referee Comment (RC1) · Anonymous Referee #2 · 5 Jan 2018

The manuscript "Hygroscopic growth study in the framework of EARLINET during the SLOPE I campaign: synergy of remote sensing and in-situ instrumentation" focuses on the aerosol hygroscopic properties as inferred from remote sensing instrumentation. The advantage of such a system is that the aerosol particles are measured directly as they are in the atmosphere without the need of any pre- or aftertreatment. In this particular study the shortcomings of traditional collated radio sounding (RS) measurements, suffering from low temporal resolution, are overcome by using a Raman lidar to measure water vapor mixing ratio profiles combined with a microwave radiometer to

retrieve temperature profiles. Additionally, a comparison was performed with RS measurements and in-situ data recorded at a thigh altitude station in the Sierra Nevada. I recommend the paper for publication in ACP after the following comments have been addressed:

Major comment:

In general, the paper presents an interesting study comparing direct measurements at different elevations with in-situ data at a nearby located mountain site. There is however a lack of information on the in-situ data. Did the ACSM and Aethalometer have a specific inlet (PM10, PM1...)? Did they share an inlet? What was the mean or median size distribution during the selected case studies? Did the APS see particles above 2-3 $\mu$m, which is mentioned to be the upper limit of the Lidar? Can dust particles be excluded during the case studies (which could lead to a lower backscatter coefficient but cannot be measured by either ACSM or Aethalometer)? It is mentioned that it is difficult to assess the uncertainties in the in-situ data, but can any upper or lower limits be estimated? Also, the size at which the hygroscopic growth factors listed in Table 1 were measured should be mentioned and a short comment on the mean/median size measured during the cases in this study should be added. Additionally, the introduction would profit from some information on the used method to retrieve in-situ hygroscopicity values (more comments on this are presented later).

General comments:

P2, line 30: Please make the section on commonly used hygroscopicity measurements clearer. The HTDMA has only been employed on the ground, whereas there are other instruments for airborne measurements (DASH-SP, WHOPS, AMS+Aethalometer...). Also add some information on the method used in this article and some pros and cons as stated for the HTDMA.

P6, line 6: Please add if any corrections were applied to the Aethalometer data and which MAC value was used to convert the absorption coefficient to the eBC concentrations (and reference). Figures: Please use the descriptions of a) b) c) ... in the text to refer to certain parts of the figures as this facilitates to follow the discussion. Also add legends to Figure 1 and mention what the $\gamma$ values given in the legends are (the text states only "sold lines").

Title of section 4.2.2: "measured and modelled f$\beta\lambda$(RH)" is a little misleading as no modelling (except HYSPLIT) was performed. Maybe rather use "measured f$\beta\lambda$(RH) and calculated using in-situ data and Mie theory" or something similar.

Figure 5: The "measured" data exhibits some kind of jump at RH=95%. Can you comment on this and why, possibly, it is not seen in the Mie calculations with the in-situ data?

Specific comments:

P3, line 26: change "on one hand" to "on the one hand"

P4, line 5 and following: please rephrase the second part of the sentence with the case of RH>60%

P4, line 23: please specify "incomplete overlap"

P6, line 6: change "werer" to "were"

P7, line 18: which GDAS resolution (degrees) was used?

P13, line 6: missing space between "similar" and "$\gamma$"

P13, line 8: change to "in one of their case studies"

P13, line 24: what does the 4% refer to exactly?

P13, line 29: change to "associated with..."

P13, line 31: change to "reported in.."

P14, line 22: use past tense i.e. "were" instead of "are"

P14, line 28: change "it is concluded" to "we concluded"

P15, line 5: change "gamma parameter" to "$\gamma$ parameter"

P15, line 6: please add a more precise description of what the 13% and 10% describe

P15, line 7: change "those" to "these"

P15, line 7-8: explain what "favorable" means; change "no-advection" to "in absence of advected air masses" or something similar

P15, line 11: change to "making it possible"

P15, line 12: change "those" to "these" or "such"

---

## Referee Comment (RC2) · Anonymous Referee #1 · 9 Feb 2018

Hygroscopic growth studies with lidar are a valuable contribution to the field of research as they are operating at ambient conditions and allow us to study lofted aerosol layers and not only the situation near the ground. The accurate profile of the relative humidity is an important issue to tackle for this kind of studies. The present study uses a microwave radiometer for the temperature profile and the water vapor Raman channels of the lidar for the water vapor mixing ratio to provide the relative humidity with a high temporal resolution. This method is compared to a study where RH is used from a radiosonde only. It is a beginning, but the authors are in risk to miss the chance to

discuss the topic comprehensively. Therefore I have two major points to include in the manuscript (see more details below):

I. Take a modeled temperature profile, preferably from GDAS (Global Data Assimilation System), to compare your results obtained by the microwave radiometer (and the radiosonde). If it agrees well, this would extend tremendously the application of your method.

II. Try to deliver extinction enhancement or even scattering enhancement factors as they are reported frequently in literature. Use all your information from the lidar and the in situ sampling to give at least an estimate. Just reporting the backscatter enhancement factor limits the outreach of your study.

Considering the following comments, I recommend the manuscript for publication after major revisions.

Mayor comments

1. The key facts should be included in the abstract. At the moment, the abstract is too descriptive, and too few results are presented. For someone who only reads the abstract, the main findings should shortly be presented.

Here or in the introduction, you should mention that the method is tested against the results of Granados-Muñoz, AMT 2015.

2. The introduction is not well written. The main structure of an introduction should be clearer:

Why is it important?

What has been done in this field?

What questions remain open?

What is your contribution?

[Figure]

Furthermore, it is not clear, which aerosols are observed and how the publication is structured. Recent literature from the studied field should appear in the introduction. Some references are given later in the manuscript, but they could already appear in the introduction. Please revise carefully the literature.

3. In general, the reader is interested in which aerosols you are observing. Very late, in Table 1, you give an overview. Do you consider it as continental aerosol, urban pollution or biogenic particles? Please add a discussion about the aerosol type. In Granados-Muñoz, AMT 2015, sulfate, marine and mineral dust particles are discussed.

4. Section 2 is not well structured: first a description of the campaign (now Sect. 2.2), then a sub section to the valley station in Granada (now Sect. 2.1) and then a sub section to the hill station with the in situ instrumentation , where you explain which instrument measures which quantity.

5. A sketch of the location would be nice (maybe a vertical cut, showing the orographic profile with valley and hill station and the distance between them). And you mention several in-situ stations at different heights, an interesting fact, that is not used later on.

6. It is great to have a station on a mountain, almost 2000 m above the lidar and several stations on the mountain slope. Somewhere you should mention hygroscopic studies which compared remote sensing measurements with (meteorological) tower based in situ instrumentation, which only reach up to approximately 200 m above ground or the use of horizontal pointing remote sensing instruments to ground-based in situ observations; and the advantage of having a mountain slope for performing such experiments.

7. Another key point, you retrieve backscatter enhancement factors and you state, that it is difficult to compare them to values found in the literature. With your Raman signals, you can retrieve the extinction coefficient and determine the extinction enhancement factor or at least the lidar ratio. For the extinction enhancement factor, there are much more literature values to compare. Eventually, your in situ measurements allow you to determine the single scattering albedo and then you can derive the scattering enhancement factor, which can be compared to results obtained by in situ observations of hygroscopic growth. These conversions would add a lot of value to the paper.

8. As I understand it right, one idea of the paper is to perform hygroscopic growth studies using a calibrated Raman lidar system without having a radiosonde available. The Raman lidar delivers the aerosol properties and water vapor mixing ratio (if the calibration constant is known). In order to derive the relative humidity, the temperature profile is needed, which you derive from the microwave radiometer. Another option would be to use the temperature profile of the GDAS model output, as there are all available radiosonde ascends are included. I would like to see, how this even easier method compares to your results.

9. p4, l6-12 What about marine aerosol in Granada?

10. p6, l25 – p7, l3 At which temporal resolution do you derive the temperature profiles and the RH profiles?

11. p12, l6-21 and Fig. 4 Why the third moment of the vertical velocity and not the vertical wind velocity is shown? The vertical velocity would give valuable information about updrafts and downdrafts.

12. p14, l1-4 The statement is not convincing and needs more explanation. You can use a particle size distribution from the mountain station to show the influence of the large particles and how frequent they are. Furthermore you can use the 1064 nm backscatter to be more sensitive for the large particles.

13. Why don't you use the backscatter at 1064 nm? In Fig. 3 and 5 you can extend your study to include the near infrared. It would add value to your publication.

Minor comments:

1. Comma instead of dot in the list of affiliations (6 times)

2. The space after a symbol or a bracket is often missing throughout the manuscript.

3. Indices should not be written in italics, except in the sum formula (Eq. (4))

4. Units should not be written in italics.

5. Maybe you should consider slightly reducing the number of abbreviations to make the paper easier to read.

6. "upward wind", better "upward wind velocity" throughout the manuscript.

7. In recent years the term ice nucleating particles (INP) is used for the aerosols which nucleate ice, see Vali et al., ACP 2015 (just for your information).

Vali, G., DeMott, P. J., Möhler, O., and Whale, T. F.: Technical Note: A proposal for ice nucleation terminology, Atmos. Chem. Phys., 15, 10263-10270, https://doi.org/10.5194/acp-15-10263-2015, 2015.

8. p4, l16-19 The instrument description is confusing, better "It emits laser pulses at . . ., and it receives backscattered photons at . . . in . . . mode"

9. p4, l21-22 What is the approximate overlap height of the system?

10. p9, Eq 4, What about rho? It is the density of which part?

11. p9, l31 and Fig. 1 Where does the lidar ratio of 65 sr come from?

12. p11, l26 "fine/coarse predominance" better "size"

13. p11, l28 "predominance of coarse particles", better "predominance of larger particles"

14. p13, l26-27 Please indicate the uncertainty ranges for the 4 derived backscatter enhancement factors as it is done in the conclusion.

15. p14, l4 Please repeat the horizontal distance at this point.

16. Tab. 3 In the caption, be consistent with the date: 16th June 2016

17. Fig. 1+3 units should not be written in italics, see beta (. . .)

18. Fig. 1 d+h It is difficult to separate points and lines.

19. Fig. 2 Height range up to 4 or 6 km is sufficient to show. Could you please state (in the caption) the time of sunset as additional information?

20. Fig. 3 in the caption: backscatter at 1064 nm is not shown, but mentioned.

21. Fig. 4 It would be better to just show the same time interval as in Fig. 2 (1700-0000 UTC) to increase the number of details.

22. References:

- Kotchenruther et al., 1999 (not 1998)

- List, 1951, strange "f&"

- p13,l 17 no Titos et al. (2014b) only Titos et al. (2014)

---

## Author Comment (AC1) · 27 Mar 2018

Major comment:

In general, the paper presents an interesting study comparing direct measurements at different elevations with in-situ data at a nearby located mountain site. There is however a lack of information on the in-situ data.

We agree with the referee#2 that there is a lack of information on the in-situ instrumentation and its data treatment. Accordingly, we have extended the "Experimental

site and instrumentation" section including a detailed description of the measurement set-up as follows:

p7, line 3 - 16 "At Sierra Nevada station (SNS, 37.09 N, 3.38° W, 2500 m a.s.l.), state-of-the-art in-situ instrumentation was operated to characterize aerosol properties. The inlet at SNS station is a whole air inlet located in the rooftop of a 3-story building. It is made up of stainless steel tube, with dimensions of 10 cm in diameter and 2.5 m in length. Inside the main tube there is a laminar flow of 100 litres per minute and there are several stainless steel pipes that drive the sampling air to the different instruments. Each one of the stainless steel pipes extracts the appropriate flow for each instrument. Different diameters of the pipes have been selected in order to optimize the efficiency of the system [Baron and Willeke, 2001]. The instrumentation used in this study includes an Aerodyne Aerosol Chemical Speciation Monitor (ACSM, Aerodyne Research Inc.), an Aethalometer (AE33 model, Magee Scientific, Aerosol d.o.o.), an Aerodynamic Particle Sizer (APS, TSI 3321) spectrometer and a Scanning Mobility Particle Sizer (SMPS, TSI 3938) spectrometer; all of them connected to the main inlet. The ACSM was used to measure on-line submicron inorganic (nitrate, sulphate and ammonium) and organic aerosol (OA) concentrations. Equivalent black carbon, eBC, mass concentration was obtained from measurements of the Aethalometer AE33 at 880 nm. A mass absorption cross section of 7.77 m2g-1 was used to convert the absorption coefficients at 880 nm in eBC mass concentrations (Drinovec et al., 2014). Particle number size distributions were retrieved by a combination of the measurements performed with the SMPS in the diameter range 13-600 nm and the APS for the range 0.6-20 $\mu$m."

Did the ACSM and Aethalometer have a specific inlet (PM10, PM1. . .)? Did they share an inlet? Yes, both instruments share the same inlet. It is a whole air inlet located in the rooftop of the building. For further details on the inlet and experimental setup see our comment above.

What was the mean or median size distribution during the selected case studies?

From the Fig. 1 R2 we can see the mean size distribution during the hygroscopic case investigated in this manuscript. The number size distribution has two main peaks at 35 and 115 nm.

Did the APS see particles above 2-3 $\mu$m, which is mentioned to be the upper limit of the Lidar?

From the mean size distribution (Figure 1 R2) during the hygroscopic case it can be observed that most of the aerosol is in the fine mode (< 1 $\mu$m). The upper limit of the lidar is not 2-3 $\mu$m, in the manuscript we stated that due to the lidar's wavelengths the instrument is more efficient at those wavelengths. Therefore, to avoid confusion, the following paragraph has been rephrased,

p14, line 7 - 9 "At SNS station, according to the mean size distribution calculated with SMPS during the hygroscopic growth case two main peaks were detected at 35 and 115 nm (from 20:00 to 21:00 UTC), therefore, it can be observed that most of the aerosol is in the fine mode (< 1 $\mu$m), but the lidar's wavelengths used are more efficient at those wavelengths."

Can dust particles be excluded during the case studies (which could lead to a lower backscatter coefficient but cannot be measured by either ACSM or Aethalometer)?

During this campaign, the chemical composition was measured with the ACSM and the Aethalometer. This set-up does not allow us to determine the mass concentration of dust particles neither to detect their presence. The air mass trajectories arriving at SNS during 16 June 2016 were coming from the North Atlantic Ocean and the atmospheric situation during this day and the previous days was characterized by clean conditions. Considering also additional data gathered during the campaign at SNS we can conclude that dust particles were not present during the case investigated here. In this sense, the scattering Angstrom Exponent (SAE) can be used as an indicator of the predominance of fine (SAE around 2) or coarse particles (SAE < 1). From the scattering coefficients measured with a nephelometer (TSI 3563) at SNS station, the

calculated SAE was > 1.2 during the hygroscopic case under study. In addition, AOD-derived Angstrom Exponent from AERONET Level 2.0 were above 1.4 during the whole day (16 June 2016). The fine mode fraction retrieved by AERONET was above 0.83 evidencing that coarse particles did not predominate.

It is mentioned that it is difficult to assess the uncertainties in the in-situ data, but can any upper or lower limits be estimated?

The uncertainty in the f (RH) calculation is a function of the individual measurement uncertainties in the particle number size distribution and in the chemical composition. Furthermore, the different assumptions made to calculate the backscattering coefficients within the Mie model contribute to this uncertainty as well. These assumptions are: Since no experimental measurements of the growth factor, g (RH), were available during the campaign, the g (RH) was calculated based on g(RH) from the literature for the individual chemical components measured with the ACSM+Aethalometer. The change in the size distribution with changing RH is calculated based on a constant and size-independent g (RH). The change in refractive index with changing RH is calculated from a volume-weighting of the aerosol dry refractive index and the refractive index of water, using the mean g (RH) (Hale and Querry, 1973).

This procedure has been extensively used and it has been proven to provide successful results in closure studies (e.g. Fierz-Schmidhauser et al., 2010ACP; Zieger et al., 2013). Concerning the uncertainties in the retrieval, Adam et al. (2012) calculated f (RH) using Mie theory and performed a sensitivity study concluding that errors in the backscattering enhancement factor can vary from 10 up to 30% as the RH changes from <50% to 90%. This information has been included in the revised manuscript.

The following information is now included in the manuscript: p9, line 11 - 17 "Estimations of ãĂŰ fãĂŮ_($\beta$ ) ($\lambda$,RH) uncertainty are very scarce because of their high complexity. Some studies (e. g. Adam et al., 2012; Zieger et al., 2013) provided estimations based on sensitivity analysis using Mie model calculations, reporting errors around 20% f_$\sigma$ ($\lambda$,RH). where $\sigma$ is the scattering coefficient. Titos et al. (2016) reported uncertainty estimations based on Monte-Carlo techniques, concluding that the more hygroscopic the aerosol, the higher is the uncertainty in f_$\sigma$ ($\lambda$,RH) especially at high RH (RH > 80%). For moderate-hygroscopic aerosol, it was established a lower limit for the uncertainty in f_$\sigma$ ($\lambda$,RH) of around 30-40% using nephelometry techniques."

Also, the size at which the hygroscopic growth factors listed in Table 1 were measured should be mentioned and a short comment on the mean/median size measured during the cases in this study should be added.

We have included the diameters in the caption of Table 1 and a short comment on the mean size distribution, as suggested by the reviewer.

p14, line 6 – 9 "At SNS station, according to the mean size distribution calculated with SMPS during the hygroscopic growth case two main peaks were detected at 35 and 115 nm (from 20:00 to 21:00 UTC), therefore, it can be observed that most of the aerosols are in fine mode (< 1 $\mu$m), but the lidar's wavelengths used are more efficient at those wavelengths."

Additionally, the introduction would profit from some information on the used method to retrieve in-situ hygroscopicity values (more comments on this are presented later).

We agree with the referee#2. A detail response is given in the next comment.

p2, line 24 "Several studies have been carried out over the past years in order to evaluate how water uptake affects aerosol properties. One parameter used to quantify these changes is the so-called aerosol hygroscopic enhancement factor, f ($\lambda$,RH), defined as the ratio between aerosol optical/microphysical properties at wet atmospheric conditions and the corresponding reference value at dry conditions (Hänel 1976; Ferrare et al. 1998; Feingold et al., 2003; Veselovskii et al., 2009; Granados-Muñoz et al., 2015; Titos et al., 2014, 2016, and references therein). Most of the previous studies investigating aerosol hygroscopicity are based on in-situ measurements. One of the most commonly used in-situ instruments for measuring aerosol hygroscopicity is the Humidified Tandem Differential Mobility Analyzer (HTDMA) (e.g. Swietlicki et al., 2008) that measures the hygroscopic growth factor g(RH) that quantifies the change in particle diameter due to water uptake. Humidified tandem nephelometers have been extensively used as well to quantify the effect of the hygroscopic growth in the aerosol optical properties, namely scattering, backscattering and extinction coefficients (e.g., Pilat and Charlson 1966; Titos et al., 2016). There are other in-situ instruments such as the white-light humidified optical particle spectrometer (WHOPS) (Rosatti et al., 2015) or the Differential Aerosol Sizing and Hygroscopicity Spectrometer Probe (DASH-SP), (Sorooshian et al., 2008) that have been used to determine the impact of enhance RH on the aerosol properties from airborne platforms. The effect of RH on the aerosol optical properties can be also determined with Mie model calculations (e.g. Adams et al., 2012; Fierz-Schmidhauser et al., 2010; Zieger et al., 2013) using the measured size distribution and chemical composition as inputs. For this calculation, the g (RH) is also needed as input. This factor can be determined experimentally (using HTDMA measurements for example) or it can be inferred from the individual growth factors of the different chemical compounds. The assumption of some aerosol properties such as the refractive index or the growth factor based on the chemical composition is the main drawback of this method. Generally speaking, one important limitation of most in-situ techniques is that they modify the ambient conditions and are also subject of particles losses in the sampling lines, therefore altering the real atmospheric aerosol properties."

General comments:

P2, line 30: Please make the section on commonly used hygroscopicity measurements clearer. The HTDMA has only been employed on the ground, whereas there are other instruments for airborne measurements (DASH-SP, WHOPS, AMS+Aethalometer...). Also add some information on the method used in this article and some pros and cons

as stated for the HTDMA.

According to the referee#2's suggestion we have clarified this. We have explicitly mentioned the different instruments used to investigate aerosol hygroscopicity from the ground and from airborne platforms but we have not extended the discussion much further since a review of techniques is not the focus of this manuscript. These lines have been replaced in the manuscript by. For detailed information see the comment above.

P6, line 6: Please add if any corrections were applied to the Aethalometer data and which MAC value was used to convert the absorption coefficient to the eBC concentrations (and reference).

No corrections were applied to the Aethalometer AE33 data and the MAC used was 7.77 m2g-1, as recommended by the manufacturer for this wavelength (Drinovec et al., 2014). This information has been added to the instrumentation section.

p7, line 13 – 16 "Equivalent black carbon, eBC, mass concentration was obtained from measurements of the Aethalometer AE33 at 880 nm. A mass absorption cross section of 7.77 m2g-1 was used to convert the absorption coefficients at 880 nm in eBC mass concentrations (Drinovec et al., 2015)"

Figures: Please use the descriptions of a) b) c) . . . in the text to refer to certain parts of the figures as this facilitates to follow the discussion. Also add legends to Figure 1 and mention what the $\gamma$ values given in the legends are (the text states only "sold lines"). We agree with the referee#2. These changes have been performed in the manuscript.

Title of section 4.2.2: "measured and modelled f$\beta\lambda$(RH)" is a little misleading as no modelling (except HYSPLIT) was performed. Maybe rather use "measured f$\beta\lambda$(RH) and calculated using in-situ data and Mie theory" or something similar

Following the referee#2's suggestion, we have decided to modify the title as it is follow:

P13, line 29 "f_$\beta^{\lambda}$ (RH) measured and retrieved by combining in-situ data and Mie

theory"

Figure 5: The "measured" data exhibits some kind of jump at RH=95%. Can you comment on this and why, possibly, it is not seen in the Mie calculations with the in-situ data?

The jump seen on figure 5 on measured data, could be associated with the uncertainties on the RH calculation itself mostly above RH>95%, because of the combination of MWR temperature and calibrated WVMR signal. On this hand, the RH profile could have some data tendency to keeping constant meanwhile backscatter coefficient could remain increasing above RH=95 %, which may cause to the final calculation an increase of the uncertainty of f (RH).

Specific comments:

P3, line 26: change "on one hand" to "on the one hand" Done P4, line 5 and following: please rephrase the second part of the sentence with the case of RH>60%

We have rephrased it in the manuscript:

p5, line 6 - 8 "Navas-Guzmán et al. (2014) analyzed one year of measurements of RH profiles at Granada showing that RH values are low (below 60%) in the 75% of the cases studied within 1.0 and 2.0 km a.s.l. This study also showed that most of the cases with RH above 60% are found in spring and winter seasons."

P4, line 23: please specify "incomplete overlap"

We have included the following sentence in the manuscript:

p5, line 29-30 "Atmospheric information retrieved from lower regions is limited by the full overlap height, which is reached above 1.3 km a.s.l due to the system configuration (Guerrero-Rascado et al., 2010; Navas-Guzmán et al., 2011)"

P6, line 6: change "werer" to "were" Done P7, line 18: which GDAS resolution (degrees) was used?

We have included in the manuscript:

p8, line 22 - 23 "GDAS meteorological data used have a spatial resolution of 0.5°x0.5° available since 2010 with daily files every three hours on the native GFS hybrid sigma coordinate system."

P13, line 6: missing space between "similar" and "$\gamma$" Done P13, line 8: change to "in one of their case studies" Done P13, line 24: what does the 4% refer to exactly?

It means that taking as the reference $f_\beta\hat{}355$ and $f_\beta\hat{}532$ from Mie calculations and making the relative error calculation with measured values for each wavelength, this relative error remains below 5 %. The relative error for $f_\beta\hat{}355$=2.7 % and relative error for $f_\beta\hat{}532$=4.3 %. P13, line 29: change to "associated with. . ." Done P13, line 31: change to "reported in.." Done P14, line 22: use past tense i.e. "were" instead of "are" Done P14, line 28: change "it is concluded" to "we concluded" Done P15, line 5: change "gamma parameter" to "$\gamma$ parameter" Done P15, line 6: please add a more precise description of what the 13% and 10% describe We have commit an error in the percentages, to be precise, we found good agreement on $\gamma$ parameters measured (ãĂŰ $\gamma$ãĂŮ$\hat{}532$=0.48±0.01 and $\gamma\hat{}355$=0.40±0.01) respect to Mie calculated ones (ãĂŰ $\gamma$ãĂŮ$\hat{}532$=0.53±0.02 and $\gamma\hat{}355$=0.45±0.02 ), with relative differences between measured and calculated of 9 % at 532 nm and 11 % at 355 nm, taking the calculated values as reference. P15, line 7: change "those" to "these" Done P15, line 7-8: explain what "favorable" means; change "no-advection" to "in absence of advected air masses" or something similar Favorable refers to the good atmospheric conditions needed to evaluate these phenomena like vertical homogeneity (good mixed layers), same origin of the aerosols within the analyzed layer, low horizontal velocity and the absence of advected air masses into the evaluated column. P15, line 11: change to "making it possible" Done P15, line 12: change "those" to "these" or "such" Done

[Figure]

**Fig. 1.** Figure 1 R2. Mean particle number size distribution during the hygroscopic growth case on 16 June from 20:00 to 21:00 UTC.

---

## Author Comment (AC2) · 28 Mar 2018

Author's response

We thank the anonymous reviewers for his/her comments and suggestions that have helped to improve the quality of the manuscript. According to the referees' reports, the following changes has been performed on the original manuscript and a point-by-point response is included below,

Answers to Referee#1:

[Figure]

Two major points to include in the manuscript: I. Take a modelled temperature profile, preferably from GDAS (Global Data Assimilation System), to compare your results obtained by the microwave radiometer (and the radiosonde). If it agrees well, this would extend tremendously the application of your method.

Following the reviewer#1's suggestion, Fig. 1R1 (a). presents a comparison among microwave radiometer (MWR), radiosounding (RS) and GDAS relative humidity data on 22th Jul 2013 at 20:00 UTC, selected as an example of one studied case in the manuscript. The following configuration is used in order to get a dataset comparable in time: a time-averaged profile from 20:00 to 21:00 UTC for MWR, GDAS output at 21:00 UTC, the RS launched at 20:00 UTC. This figure shows a negative bias for RHlidar+MWR within 0 to 10 % for almost all profile, instead the variability between RHlidar+GDAS is higher from -15 % to almost 20 % in the upper profile.

The disagreement between GDAS and RS profiles is mainly associated to two factors: (i) the complex terrain where the measurement station is located, surrounded by mountains of high elevation (up to more than 3000 m a.s.l in a very short horizontal distance of few tenths of kilometres) that makes more difficult for models to provide accurate thermodynamics profiles for this location; (ii) GDAS profiles have a lower temporal resolution (3 h) than the MWR one, which gives temperature profiles each 2 min and here is averaged up to 1 h. The combination of these two factors is the reason why we conducted our study in terms of MWR data, although we agree with the reviewer that the use of GDAS temperature profiles would extend the applicability of the methodology presented in this manuscript for locations with less complex orography.

Try to deliver extinction enhancement or even scattering enhancement factors as they are reported frequently in literature. Use all your information from the lidar and the in situ sampling to give at least an estimate. Just reporting the backscatter enhancement factor limits the outreach of your study.

We agree with the fact that the literature mostly reports values about scattering and

extinction enhancement factors. It is due that most of hygroscopic studies have been addressed using in-situ instrumentation. The optical parameters that can be obtained from lidar measurements are backscatter (scatter at $180°$) and extinction coefficients. Unfortunately, the extinction retrievals for the presented case are not available due to the low quality of the Raman-shifted lidar signals, and this is the reason why we have only focused on the backscatter coefficient. However, the outreach of our results is very important because most of the ground-based and satellite lidar observations provide only vertical information of the backscatter coefficient. Mayor comments

The key facts should be included in the abstract. At the moment, the abstract is too descriptive, and too few results are presented. For someone who only reads the abstract, the main findings should shortly be presented. Here or in the introduction, you should mention that the method is tested against the results of Granados-Muñoz, AMT 2015.

Following the recommendations of the referee#1, we have included in the abstract the following lines in blue colour:

"This study focuses on the study of aerosol hygroscopic growth during Sierra Nevada Lidar AerOsol Profiling Experiment (SLOPE I) campaign by using the synergy of active and passive remote sensors at Granada valley station and in-situ instrumentation at a mountain station (Sierra Nevada). To this end, a methodology based on the combination of calibrated water vapour mixing ratio (r) profiles, retrieved from an EARLINET multiwavelength Raman lidar (RL), and continuous temperature profiles from a microwave radiometer (MWR) for obtaining relative humidity (RH) profiles with high temporal resolution is used. This methodology is validated against an approach using radiosounding (RS) data, obtaining differences in hygroscopic growth parameter ($\gamma$) lower than 5% between the methodology based on RS and that based on remote sensing. During SLOPE I the remote sensing methodology used for aerosol hygroscopic growth studies has been checked against Mie calculations of hygroscopic growth using in-situ measurements of particle number size distribution measured at SNS. The hygroscopic case observed during SLOPE I showed an increase in particle backscatter coefficient at 355 and 532 nm with the relative humidity (with RH ranging between 78-98%), but also a decrease in backscatter-related Ångström exponent (AE) and particle linear depolarization ratio (PLDR) indicating that the particle became larger and more spherical due to hygroscopic processes.Vertical and horizontal wind analysis is performed by means of a co-located Doppler lidar system at IISTA-CEAMA station, in order to evaluate the horizontal and vertical dynamics of the air masses. Finally, the Hänel parameterization is applied to experimental data for both stations and we found good agreement on $\gamma$ parameters measured (ãĂŰ $\gamma$ãĂŮˆ( 532)=0.48±0.01 and ãĂŰ$\gamma$ ãĂŮˆ355=0.40±0.01) respect to calculated (ãĂŰ $\gamma$ ãĂŮˆ532=0.53±0.02 and $\gamma$ˆ( 355)=0.45±0.02 ), with relative differences between measured and calculated up to 9 % at 532 nm and 11 % at 355 nm."

The introduction is not well written. The main structure of an introduction should be clearer: Why is it important? What has been done in this field? What questions remain open? What is your contribution?

Discussion paper Furthermore, it is not clear, which aerosols are observed and how the publication is structured. Recent literature from the studied field should appear in the introduction. Some references are given later in the manuscript, but they could already appear in the introduction. Please revise carefully the literature.

Following the reviewer#1's suggestions 2 and 3, the introduction has been restructured as follows

[revised manuscript text omitted]

In general, the reader is interested in which aerosols you are observing. Very late, in Table 1, you give an overview. Do you consider it as continental aerosol, urban pollution or biogenic particles? Please add a discussion about the aerosol type. In GranadosMuñoz, AMT 2015, sulfate, marine and mineral dust particles are discussed.

In order to improve the discussion about aerosol type we have added the following paragraph.

P14, line 14 -18 "Thus, the predominant aerosol studied during the event is a composition of smoke and urban polluted aerosol. This assumption about the aerosol type is supported by the relatively high sulphate concentration observed at SNS and the results discussed in section 4.2.1 (lidar properties, backward trajectories analyses). The backscatter-related AE values (close to 1.5) indicates a predominance of fine particles which is connected with the presence of smoke over that comes from North America as suggested by the backward trajectories analyses."

Section 2 is not well structured: first a description of the campaign (now Sect. 2.2), then a sub section to the valley station in Granada (now Sect. 2.1) and then a sub section to the hill station with the in-situ instrumentation, where you explain which instrument measures which quantity.

We have re-structured this section of the manuscript following the reviewer instructions, as follows: 2.1 SLOPE I field campaign 2.2 IISTA-CEAMA station 2.3 Sierra Nevada Station

A sketch of the location would be nice (maybe a vertical cut, showing the orographic profile with valley and hill station and the distance between them). And you mention several in-situ stations at different heights, an interesting fact, that is not used later on.

We agree with the reviewer and therefore we have added a new figure with the orographic profile showing the location of the IISTA-CEAMA station (located in the valley) and SNS station (located in the slope of Sierra Nevada, at 2500 m asl). During this

campaign, only one in-situ station was operative so we have modified the manuscript accordingly.

It is great to have a station on a mountain, almost 2000 m above the lidar and several stations on the mountain slope. Somewhere you should mention hygroscopic studies which compared remote sensing measurements with (meteorological) tower based in situ instrumentation, which only reach up to approximately 200 m above ground or the use of horizontal pointing remote sensing instruments to ground-based in situ observations; and the advantage of having a mountain slope for performing such experiments.

We have added some information in the introduction about works related with aerosol hygroscopic growth and fog detection by using ceilometer attenuated backscatter data combined with instrumented in-situ tower. These works have centred their attention on forecasting, but also, they open new possibilities for low heights aerosol hygroscopicity growth studies. Also, some studies performed by synergy of Raman lidar, in-situ and MAX-DOAS instrumentation making possible a good extrapolation the extinction coefficient to the ground:

p3, line 17-23 "This aim has also been studied by Zieger et at. (2011), showing the capability to combine Raman lidar, in-situ and MAX-DOAS instrumentation for study hygroscopic growth in ambient conditions extrapolating extinction coefficient from lidar to the ground studies. Also, some studies have been performed by using Automatic Lidar and Ceilometers (ALC) for hygroscopic and fog studies mostly for forecasting purposes of fog events, through the combination of attenuated backscatter with in-situ data from instrumented towers which reach almost 200 m above ground level (Haeffelin et al., 2016). These works are focused in enhancing the possibility to study aerosol hygroscopicity growth at low levels in ambient conditions, but in this work, we are adding a comparison between ground city station to high mountain station in order to connect effects of the city over mountain and also avoid technical issues like lidar overlap."

Another key point, you retrieve backscatter enhancement factors and you state, that it

is difficult to compare them to values found in the literature. With your Raman signals, you can retrieve the extinction coefficient and determine the extinction enhancement factor or at least the lidar ratio. For the extinction enhancement factor, there are much more literature values to compare. Eventually, your in-situ measurements allow you to determine the single scattering albedo and then you can derive the scattering enhancement factor, which can be compared to results obtained by in situ observations of hygroscopic growth. These conversions would add a lot of value to the paper.

As we explain above, unfortunately the extinction retrievals for the presented case are not available due to the low quality of the Raman-shifted lidar signals, and this is the reason why we cannot perform the calculations proposed by referee#1, because of that we are not including the in-situ single scattering albedo to derive scattering enhancement factor. In this work we have only focused on the backscatter coefficient, but the outreach of our results is very important because most of the ground-based and satellite lidar observations provide only vertical information of the backscatter coefficient.

As I understand it right, one idea of the paper is to perform hygroscopic growth studies using a calibrated Raman lidar system without having a radiosonde available. The Raman lidar delivers the aerosol properties and water vapor mixing ratio (if the calibration constant is known). In order to derive the relative humidity, the temperature profile is needed, which you derive from the microwave radiometer. Another option would be to use the temperature profile of the GDAS model output, as there are all available radiosonde ascends are included. I would like to see, how this even easier method compares to your results.

As we shown on Fig. 1R1, we have compared MWR and GDAS temperature data profiles with RS temperature profiles, so the results were not good enough. The main argument is that Granada is located in a very complex terrain because of the Sierra Nevada, therefore GDAS meteorological data doesn't fit enough good and also the temporal resolution of the GDAS data (3h) is quite lower compared with MWR. The Fig. 3R1, shows the calculation of f (RH) by using temperature from MWR (lidar +

MWR) and temperature from GDAS (lidar + GDAS), in order to show as an example that relative error between $\gamma$_RS and $\gamma$_MWR is lower than 4 % instead $\gamma$_RS and $\gamma$_GDAS is lower than 30 %, taking $\gamma$_RS=0.99 as theoretical value.

10. p4, l6-12 What about marine aerosol in Granada?

We have added the following paragraph explaining the conditions with marine aerosols in Granada

p5, line 14 - 19 "The probability of marine particles to reach the city is low taking into account that Granada is far away from the coast about 50 km in straight line, the marine particles would have to overpass some mountains in the path from the sea to the city and the air masses monitored over Granada are really dry. Also, Titos et al. (2014) showed that the contribution of marine aerosols to PM10 mass concentration was almost negligible (<3%) at IISTA-CEAMA station during the period 2006-2010. In addition, this work also refers to the identification of fine (PM1) and coarse (PM10) particulate matter in an urban environment of Granada".

11. p6, l25 – p7, l3 At which temporal resolution do you derive the temperature profiles and the RH profiles?

We have added the following information on the manuscript:

p8, line 1 -3 'Temperature profiles from the MWR, which are continuously measured every 2 min, combined with 30 min- averaged r (z) profiles as proposed by Navas-Guzmán et al. (2014), are used to retrieve the RH profiles required for aerosol hygroscopic growth studies each 30 min. The following equation is used for retrieve the RH profiles'

12. p12, l6-21 and Fig. 4 Why the third moment of the vertical velocity and not the vertical wind velocity is shown? The vertical velocity would give valuable information about updrafts and downdrafts.

The third moment of the vertical velocity (skewness) provide us a detailed information about the aerosols dynamic. It is directly associated with Turbulent Kinetic Energy equation, providing us information about the direction of convective movements, its intensity, and consequently if there is a predominance of updrafts or downdrafts (Moreira, et al., 2018a). Therefore, the observation of this moment together with the direction and speed of horizontal wind can provide us a detailed information about aerosol origin and density in the chosen period.

13. p14, l1-4 The statement is not convincing and needs more explanation. You can use a particle size distribution from the mountain station to show the influence of the large particles and how frequent they are. Furthermore, you can use the 1064 nm backscatter to be more sensitive for the large particles.

According to the mean size distribution (Figure 4 R1) during the hygroscopic case it can be observed that most of the aerosol particles are in the fine mode (< 1 $\mu$m), but we stated that due to the lidar's wavelengths the instrument is more efficient at those wavelengths. 14. Why don't you use the backscatter at 1064 nm? In Fig. 3 and 5 you can extend your study to include the near infrared. It would add value to your publication.

We completely agree with referee#1, however the 1064 nm channel was no operational during this phase of the campaign.

Minor comments:

1. Comma instead of dot in the list of affiliations (6 times)

Done 2. The space after a symbol or a bracket is often missing throughout the manuscript. Done

3. Indices should not be written in italics, except in the sum formula (Eq. (4))

Done

4. Units should not be written in italics.

Done

5. Maybe you should consider slightly reducing the number of abbreviations to make the paper easier to read.

Done

6. "upward wind", better "upward wind velocity" throughout the manuscript.

Done

7. In recent years the term ice nucleating particles (INP) is used for the aerosols which nucleate ice, see Vali et al., ACP 2015 (just for your information). Vali, G., DeMott, P. J., Möhler, O., and Whale, T. F.: Technical Note: A proposal for ice nucleation terminology, Atmos. Chem. Phys., 15, 10263-10270, https://doi.org/10.5194/acp-15-10263-2015, 2015. 8. Done

9. p4, l16-19 The instrument description is confusing, better "It emits laser pulses at . . ., and it receives backscattered photons at . . . in . . . mode"

We have added on the manuscript: p5, line 23-26 "It emits laser pulses at 355 and 532 nm (parallel and perpendicular polarization channels) and 1064 nm and it receives backscattered photons at 355, 532 and 1064 nm in analog and photon counting modes. Also, it collects Raman backscattered photons at 607 and 387 nm for molecular nitrogen (N2) and at 408 nm for water vapor (H2O) in photon counting mode"

10. p4, l21-22 What is the approximate overlap height of the system?

We have added this information on the manuscript: p5, line 29-30 "Atmospheric information retrieved from lower regions is limited by the full overlap height, which is reached above 1.3 km a.s.l due to the system configuration (Guerrero-Rascado et al., 2010; Navas-Guzmán et al., 2011)"

10. p9, Eq 4, What about rho? It is the density of which part? In Eq 4, rho is the total density of the aerosol, now included on the manuscript. 11. p9, l31 and Fig. 1 Where

does the lidar ratio of 65 sr come from? This lidar ratio was obtained in Granados-Muñoz et al., 2015, so we have used the same values retrieved in this work in order to be comparable. 12. p11, l26 "fine/coarse predominance" better "size" Done 13. p11, l28 "predominance of coarse particles", better "predominance of larger particles" Done 14. p13, l26-27 Please indicate the uncertainty ranges for the 4 derived backscatter enhancement factors as it is done in the conclusion. Done 15. p14, l4 Please repeat the horizontal distance at this point. Done 16. Tab. 3 In the caption, be consistent with the date: 16th June 2016 Done 17. Fig. 1+3 units should not be written in italics, see beta (. . .) We have reviewed text in the Fig. 1+3, but we have write these symbols in latex format so beta looks like italic but it is the format itself. 18. Fig. 1 d+h It is difficult to separate points and lines. Done 19. Fig. 2 Height range up to 4 or 6 km is sufficient to show. Could you please state (in the caption) the time of sunset as additional information? Done 20. Fig. 3 in the caption: backscatter at 1064 nm is not shown but mentioned. Done 21. Fig. 4 It would be better to just show the same time interval as in Fig. 2 (1700-0000 UTC) to increase the number of details. Done 22. References: - Kotchenruther et al., 1999 (not 1998) - List, 1951, strange "f&" - p13,l 17 no Titos et al. (2014b) only Titos et al. (2014) Done
* * *
[Figure]

[Figure]

**Fig. 1.** Figure 1 R1. RH comparison for 22th Jul 2013 around 20:00-21_00 UTC. (a) RH profiles retrieved from combination of lidar+MWR (black line), lidar+GDAS (blue line) and RS (red line) and (b) Bias calcula

[Figure]

**Fig. 2.** Figure 2 R1. Vertical profile of SNS. The yellow star refers to IISTA-GRANADA station and green star refers to SNS.

[Figure]

**Fig. 3.** Figure 3 R1. Backscatter enhancement factor retrieved for 22th July 2013. In red lines/dots is shown Lidar + MWR and blue lines/dots shown Lidar + GDAS calculations.

[Figure]

**Fig. 4.** Figure 4 R1. Mean particle number size distribution during the hygroscopic growth case on 16 June from 20:00 to 21:00 UTC.

---

## Author Response (AR2)

Hygroscopic growth study in the framework of EARLINET during the SLOPE I campaign: synergy of remote sensing and in-situ instrumentation (acp-2017-993)

Bedoya-Velásquez A.E.[1,2,3]., Navas-Guzmán F.[4], Granados-Muñoz M.J.[5], Titos G.[1,6], Román R.[1,2,11], Casquero-Vera J.A.[1,2], Ortiz-Amezcua P. [1,2], Benavent-Oltra J.A.[1,2], Moreira G. A.[1,2,7]., Montilla-Rosero E.[8], Hoyos C.D.[9], Artiñano B.[10], Coz E.[10], Alados-Arboledas L[1,2]and Guerrero- Rascado J.L.[1,2].

*Correspondence to*: Andrés Esteban Bedoya Velásquez (aebedoyav@correo.ugr.es)

**Author's response**

We thank the anonymous reviewers for his/her comments and suggestions that have helped to improve the quality of the manuscript. According to the referees' reports, the following changes has been performed on the original manuscript and a point-by-point response is included below, where blue colour is related with answers for referee#1 and red colour for referee#2.

**Answers to Referee#1:**

Two major points to include in the manuscript:

  **I.** **Take a modelled temperature profile, preferably from GDAS (Global Data Assimilation System), to compare your results obtained by the microwave radiometer (and the radiosonde). If it agrees well, this would extend tremendously the application of your method.**

Following the reviewer#1's suggestion, Fig. 1R1 (a). presents a comparison among microwave radiometer (MWR), radiosounding (RS) and GDAS relative humidity data on 22th Jul 2013 at 20:00 UTC, selected as an example of one studied case in the manuscript. The following configuration is used in order to get a dataset comparable in time: a time-averaged profile from 20:00 to 21:00 UTC for MWR, GDAS output at 21:00 UTC, the RS launched at 20:00 UTC. This figure shows a negative bias for $RH_{lidar+MWR}$ within 0 to 10 % for almost all profile, instead the variability between $RH_{lidar+GDAS}$ is higher from -15 % to almost 20 % in the upper profile.

The disagreement between GDAS and RS profiles is mainly associated to two factors:

(i)     the complex terrain where the measurement station is located, surrounded by mountains of high elevation (up to more than 3000 m a.s.l in a very short horizontal distance of few tenths of kilometres) that makes more difficult for models to provide accurate thermodynamics profiles for this location;

(ii)    GDAS profiles have a lower temporal resolution (3 h) than the MWR one, which gives temperature profiles each 2 min and here is averaged up to 1 h.

The combination of these two factors is the reason why we conducted our study in terms of MWR data, although we agree with the reviewer that the use of GDAS temperature profiles would extend the applicability of the methodology presented in this manuscript for locations with less complex orography.

[Figure]

[Figure]

**Figure 1R1.** RH comparison for 22[th] Jul 2013 around 20:00-21_00 UTC. (a) RH profiles retrieved from combination of lidar+MWR (black line), lidar+GDAS (blue line) and RS (red line) and (b) Bias calculation between lidar+MWR (redline), lidar+GDAS (blue line)

II.  **Try to deliver extinction enhancement or even scattering enhancement factors as they are reported frequently in literature. Use all your information from the lidar and the in situ sampling to give at least an estimate. Just reporting the backscatter enhancement factor limits the outreach of your study.**

We agree with the fact that the literature mostly reports values about scattering and extinction enhancement factors. It is due that most of hygroscopic studies have been addressed using in-situ instrumentation. The optical parameters that can be obtained from lidar measurements are backscatter (scatter at 180º) and extinction coefficients. Unfortunately, the extinction retrievals for the presented case are not available due to the low quality of the Raman-shifted lidar signals, and this is the reason why we have only focused on the backscatter coefficient. However, the outreach of our results is very important because most of the ground-based and satellite lidar observations provide only vertical information of the backscatter coefficient.

**Mayor comments**

1.  **The key facts should be included in the abstract. At the moment, the abstract is too descriptive, and too few results are presented. For someone who only reads the abstract, the main findings should shortly be presented. Here or in the introduction, you should mention that the method is tested against the results of Granados-Muñoz, AMT 2015.**

Following the recommendations of the referee#1, we have included in the abstract the following lines in blue colour:

"This study focuses on the study of aerosol hygroscopic growth during Sierra Nevada Lidar AerOsol Profiling Experiment (SLOPE I) campaign by using the synergy of active and passive remote sensors at Granada valley station and in-situ instrumentation at a mountain station (Sierra Nevada). To this end, a methodology based on the combination of calibrated water vapour mixing ratio (r) profiles, retrieved from an EARLINET multiwavelength Raman lidar (RL), and continuous temperature profiles from a microwave radiometer (MWR) for obtaining relative humidity (RH) profiles with high temporal resolution is used. This methodology is validated against an approach using radiosounding (RS) data, obtaining differences in hygroscopic growth parameter ($\gamma$) lower than 5% between the methodology based on RS and that based on remote sensing. During SLOPE I the remote sensing methodology used for aerosol hygroscopic growth studies has been checked against Mie calculations of hygroscopic growth using in-situ measurements of particle number size distribution measured at SNS. The hygroscopic case observed during SLOPE I showed an increase in particle backscatter coefficient at 355 and 532 nm with the relative humidity (with RH ranging between 78-98%), but also a decrease in backscatter-related Ångström exponent (AE) and particle linear depolarization ratio (PLDR) indicating that the particle became larger and more spherical due to hygroscopic processes.Vertical and horizontal wind analysis is performed by means of a co-located Doppler lidar system at IISTA-CEAMA station, in order to evaluate the horizontal and vertical dynamics of the air masses. Finally, the Hänel parameterization is applied to experimental data for both stations and we found good agreement on $\gamma$ parameters measured ( $\gamma^{532} = 0.48 \pm 0.01$ and $\gamma^{355} = 0.40 \pm 0.01$) respect to calculated ( $\gamma^{532} = 0.53 \pm 0.02$ and $\gamma^{355} = 0.45 \pm 0.02$ ), with relative differences between measured and calculated up to 9 % at 532 nm and 11 % at 355 nm."

2. **The introduction is not well written. The main structure of an introduction should be clearer: Why is it important? What has been done in this field? What questions remain open? What is your contribution?**

3. **Discussion paper Furthermore, it is not clear, which aerosols are observed and how the publication is structured. Recent literature from the studied field should appear in the**

**introduction. Some references are given later in the manuscript, but they could already appear in the introduction. Please revise carefully the literature.**

Following the reviewer#1's suggestions 2 and 3, the introduction has been restructured as follows

[revised manuscript text omitted]

**4. In general, the reader is interested in which aerosols you are observing. Very late, in Table 1, you give an overview. Do you consider it as continental aerosol, urban pollution or biogenic particles? Please add a discussion about the aerosol type. In GranadosMuñoz, AMT 2015, sulfate, marine and mineral dust particles are discussed.**

In order to improve the discussion about aerosol type we have added the following paragraph.

**P14, line 14 -18** "Thus, the predominant aerosol studied during the event is a composition of smoke and urban polluted aerosol. This assumption about the aerosol type is supported by the relatively high sulphate concentration observed at SNS and the results discussed in section 4.2.1 (lidar properties, backward trajectories analyses). The backscatter-related AE values (close to 1.5) indicates a predominance of fine particles which is connected with the presence of smoke over that comes from North America as suggested by the backward trajectories analyses."

**5. Section 2 is not well structured: first a description of the campaign (now Sect. 2.2), then a sub section to the valley station in Granada (now Sect. 2.1) and then a sub section to the hill station with the in-situ instrumentation, where you explain which instrument measures which quantity.**

We have re-structured this section of the manuscript following the reviewer instructions, as follows:

2.1 SLOPE I field campaign

2.2 IISTA-CEAMA station

2.3 Sierra Nevada Station

**6. A sketch of the location would be nice (maybe a vertical cut, showing the orographic profile with valley and hill station and the distance between them). And you mention several in-situ stations at different heights, an interesting fact, that is not used later on.**

We agree with the reviewer and therefore we have added a new figure with the orographic profile showing the location of the IISTA-CEAMA station (located in the valley) and SNS station (located in the slope of Sierra Nevada, at 2500 m asl). During this campaign, only one in-situ station was operative so we have modified the manuscript accordingly.

[Figure]

**Figure 2R1.** Vertical profile of SNS. The yellow star refers to IISTA-GRANADA station and green star refers to SNS.

7. **It is great to have a station on a mountain, almost 2000 m above the lidar and several stations on the mountain slope. Somewhere you should mention hygroscopic studies which compared remote sensing measurements with (meteorological) tower based in situ instrumentation, which only reach up to approximately 200 m above ground or the use of horizontal pointing remote sensing instruments to ground-based in situ observations; and the advantage of having a mountain slope for performing such experiments.**

We have added some information in the introduction about works related with aerosol hygroscopic growth and fog detection by using ceilometer attenuated backscatter data combined with instrumented in-situ tower. These works have centred their attention on forecasting, but also, they open new possibilities for low heights aerosol hygroscopicity growth studies. Also, some studies performed by synergy of Raman lidar, in-situ and MAX-DOAS instrumentation making possible a good extrapolation the extinction coefficient to the ground:

**p3, line 17-23** "This aim has also been studied by Zieger et at. (2011), showing the capability to combine Raman lidar, in-situ and MAX-DOAS instrumentation for study hygroscopic growth in ambient conditions extrapolating extinction coefficient from lidar to the ground studies. Also, some studies have been performed by using Automatic Lidar and Ceilometers (ALC) for hygroscopic and fog studies mostly for forecasting purposes of fog events, through the combination of attenuated backscatter with in-situ data from instrumented towers which reach almost 200 m above ground level (Haeffelin et al., 2016). These works are focused in enhancing the possibility to study aerosol hygroscopicity growth at low levels in ambient conditions, but in this work, we are adding a comparison between ground city station to high mountain station in order to connect effects of the city over mountain and also avoid technical issues like lidar overlap."

8. **Another key point, you retrieve backscatter enhancement factors and you state, that it is difficult to compare them to values found in the literature. With your Raman signals, you**

**can retrieve the extinction coefficient and determine the extinction enhancement factor or at least the lidar ratio. For the extinction enhancement factor, there are much more literature values to compare. Eventually, your in-situ measurements allow you to determine the single scattering albedo and then you can derive the scattering enhancement factor, which can be compared to results obtained by in situ observations of hygroscopic growth. These conversions would add a lot of value to the paper.**

As we explain above, unfortunately the extinction retrievals for the presented case are not available due to the low quality of the Raman-shifted lidar signals, and this is the reason why we cannot perform the calculations proposed by referee#1, because of that we are not including the in-situ single scattering albedo to derive scattering enhancement factor. In this work we have only focused on the backscatter coefficient, but the outreach of our results is very important because most of the ground-based and satellite lidar observations provide only vertical information of the backscatter coefficient.

9.  **As I understand it right, one idea of the paper is to perform hygroscopic growth studies using a calibrated Raman lidar system without having a radiosonde available. The Raman lidar delivers the aerosol properties and water vapor mixing ratio (if the calibration constant is known). In order to derive the relative humidity, the temperature profile is needed, which you derive from the microwave radiometer. Another option would be to use the temperature profile of the GDAS model output, as there are all available radiosonde ascends are included. I would like to see, how this even easier method compares to your results.**

As we shown on Fig. 1R1, we have compared MWR and GDAS temperature data profiles with RS temperature profiles, so the results were not good enough. The main argument is that Granada is located in a very complex terrain because of the Sierra Nevada, therefore GDAS meteorological data doesn't fit enough good and also the temporal resolution of the GDAS data (3h) is quite lower compared with MWR. The Fig. 3R1, shows the calculation of f (RH) by using temperature from

MWR (lidar + MWR) and temperature from GDAS (lidar + GDAS), in order to show as an example that relative error between $\gamma_{RS}$ and $\gamma_{MWR}$ is lower than 4 % instead $\gamma_{RS}$ and $\gamma_{GDAS}$ is lower than 30 %, taking $\gamma_{RS}$=0.99 as theoretical value.

[Figure]

**Figure 3 R1.** Backscatter enhancement factor retrieved for 22th July 2013. In red lines/dots is shown Lidar + MWR and blue lines/dots shown Lidar + GDAS calculations.

**10. p4, l6-12 What about marine aerosol in Granada?**

We have added the following paragraph explaining the conditions with marine aerosols in Granada

**p5, line 14 - 19** "The probability of marine particles to reach the city is low taking into account that Granada is far away from the coast about 50 km in straight line, the marine particles would have to overpass some mountains in the path from the sea to the city and the air masses monitored over Granada are really dry. Also, Titos et al. (2014) showed that the contribution of marine aerosols to PM10 mass concentration was almost negligible (<3%) at IISTA-CEAMA station during the period 2006-2010. In addition, this work also refers to the identification of fine (PM1) and coarse (PM10) particulate matter in an urban environment of Granada".

11. **p6, l25 – p7, l3 At which temporal resolution do you derive the temperature profiles and the RH profiles?**

We have added the following information on the manuscript:

**p8, line 1 -3** 'Temperature profiles from the MWR, which are continuously measured every 2 min, combined with 30 min- averaged r (z) profiles as proposed by Navas-Guzmán et al. (2014), are used to retrieve the RH profiles required for aerosol hygroscopic growth studies each 30 min. The following equation is used for retrieve the RH profiles'

12. **p12, l6-21 and Fig. 4 Why the third moment of the vertical velocity and not the vertical wind velocity is shown? The vertical velocity would give valuable information about updrafts and downdrafts.**

The third moment of the vertical velocity (skewness) provide us a detailed information about the aerosols dynamic. It is directly associated with Turbulent Kinetic Energy equation, providing us information about the direction of convective movements, its intensity, and consequently if there is a predominance of updrafts or downdrafts (Moreira, et al., 2018a). Therefore, the observation of this moment together with the direction and speed of horizontal wind can provide us a detailed information about aerosol origin and density in the chosen period.

**13. p14, l1-4 The statement is not convincing and needs more explanation. You can use a particle size distribution from the mountain station to show the influence of the large particles and how frequent they are. Furthermore, you can use the 1064 nm backscatter to be more sensitive for the large particles.**

According to the mean size distribution (Figure 4 R1) during the hygroscopic case it can be observed that most of the aerosol particles are in the fine mode ($< 1$ µm), but we stated that due to the lidar's wavelengths the instrument is more efficient at those wavelengths.

[Figure]

**Figure 4 R2.** Mean particle number size distribution during the hygroscopic growth case on 16 June from 20:00 to 21:00 UTC.

**14. Why don't you use the backscatter at 1064 nm? In Fig. 3 and 5 you can extend your study to include the near infrared. It would add value to your publication.**

We completely agree with referee#1, however the 1064 nm channel was no operational during this phase of the campaign.

**Minor comments:**

1. **Comma instead of dot in the list of affiliations (6 times)**

Done

2. **The space after a symbol or a bracket is often missing throughout the manuscript.**
   Done

3. **Indices should not be written in italics, except in the sum formula (Eq. (4))**

   Done

4. **Units should not be written in italics.**

Done

5. **Maybe you should consider slightly reducing the number of abbreviations to make the paper easier to read.**

Done

6. **"upward wind", better "upward wind velocity" throughout the manuscript.**

Done

7. **In recent years the term ice nucleating particles (INP) is used for the aerosols which nucleate ice, see Vali et al., ACP 2015 (just for your information). Vali, G., DeMott, P. J., Möhler, O., and Whale, T. F.: Technical Note: A proposal for ice nucleation terminology, Atmos. Chem. Phys., 15, 10263-10270, https://doi.org/10.5194/acp-15-10263-2015, 2015.**

8.

Done

9. **p4, l16-19 The instrument description is confusing, better "It emits laser pulses at . . ., and it receives backscattered photons at . . . in . . . mode"**

We have added on the manuscript:

**p5, line 23-26** "It emits laser pulses at 355 and 532 nm (parallel and perpendicular polarization channels) and 1064 nm and it receives backscattered photons at 355, 532 and 1064 nm in analog and photon counting modes. Also, it collects Raman backscattered photons at 607 and 387 nm for molecular nitrogen ($N_2$) and at 408 nm for water vapor ($H_2O$) in photon counting mode"

10. **p4, l21-22 What is the approximate overlap height of the system?**

We have added this information on the manuscript:

**p5, line 29-30** "Atmospheric information retrieved from lower regions is limited by the full overlap height, which is reached above 1.3 km a.s.l due to the system configuration (Guerrero-Rascado et al., 2010; Navas-Guzmán et al., 2011)"

10. **p9, Eq 4, What about rho? It is the density of which part?**

In Eq 4, rho is the total density of the aerosol, now included on the manuscript.

11. **p9, l31 and Fig. 1 Where does the lidar ratio of 65 sr come from?**

This lidar ratio was obtained in Granados-Muñoz et al., 2015, so we have used the same values retrieved in this work in order to be comparable.

**12. p11, l26 "fine/coarse predominance" better "size"**

Done

**13. p11, l28 "predominance of coarse particles", better "predominance of larger particles"**

Done

**14. p13, l26-27 Please indicate the uncertainty ranges for the 4 derived backscatter enhancement factors as it is done in the conclusion.**

Done

**15. p14, l4 Please repeat the horizontal distance at this point.**

Done

**16. Tab. 3 In the caption, be consistent with the date: 16th June 2016**

Done

**17. Fig. 1+3 units should not be written in italics, see beta (. . .)**

We have reviewed text in the Fig. 1+3, but we have write these symbols in latex format so beta looks like italic but it is the format itself.

**18. Fig. 1 d+h It is difficult to separate points and lines.**

Done

**19. Fig. 2 Height range up to 4 or 6 km is sufficient to show. Could you please state (in the caption) the time of sunset as additional information?**

Done

**20. Fig. 3 in the caption: backscatter at 1064 nm is not shown but mentioned.**

Done

**21. Fig. 4 It would be better to just show the same time interval as in Fig. 2 (1700-0000 UTC) to increase the number of details.**

Done

**22. References:**

**- Kotchenruther et al., 1999 (not 1998)**

**- List, 1951, strange "f&"**

**- p13,l 17 no Titos et al. (2014b) only Titos et al. (2014)**

Done

                                             **Answers to Referee#2:**

**Major comment:**

**In general, the paper presents an interesting study comparing direct measurements at different**
**elevations with in-situ data at a nearby located mountain site. There is however a lack of**
**information on the in-situ data.**

We agree with the referee#2 that there is a lack of information on the in-situ instrumentation and its data
treatment. Accordingly, we have extended the "Experimental site and instrumentation" section including
a detailed description of the measurement set-up as follows:

**p7, line 3 - 16** "At Sierra Nevada station (SNS, 37.09 N, 3.38° W, 2500 m a.s.l.), state-of-the-art in-situ
instrumentation was operated to characterize aerosol properties. The inlet at SNS station is a whole air
inlet located in the rooftop of a 3-story building. It is made up of stainless steel tube, with dimensions of
10 cm in diameter and 2.5 m in length. Inside the main tube there is a laminar flow of 100 litres per minute
and there are several stainless steel pipes that drive the sampling air to the different instruments. Each one
of the stainless steel pipes extracts the appropriate flow for each instrument. Different diameters of the
pipes have been selected in order to optimize the efficiency of the system [Baron and Willeke, 2001]. The
instrumentation used in this study includes an Aerodyne Aerosol Chemical Speciation Monitor (ACSM,
Aerodyne Research Inc.), an Aethalometer (AE33 model, Magee Scientific, Aerosol d.o.o.), an
Aerodynamic Particle Sizer (APS, TSI 3321) spectrometer and a Scanning Mobility Particle Sizer (SMPS,
TSI 3938) spectrometer; all of them connected to the main inlet. The ACSM was used to measure on-line
submicron inorganic (nitrate, sulphate and ammonium) and organic aerosol (OA) concentrations.

Equivalent black carbon, eBC, mass concentration was obtained from measurements of the Aethalometer AE33 at 880 nm. A mass absorption cross section of 7.77 $m^2g^{-1}$ was used to convert the absorption coefficients at 880 nm in eBC mass concentrations (Drinovec et al., 2014). Particle number size distributions were retrieved by a combination of the measurements performed with the SMPS in the diameter range 13-600 nm and the APS for the range 0.6-20 µm."

**Did the ACSM and Aethalometer have a specific inlet (PM10, PM1. . .)? Did they share an inlet?**

Yes, both instruments share the same inlet. It is a whole air inlet located in the rooftop of the building. For further details on the inlet and experimental setup see our comment above.

**What was the mean or median size distribution during the selected case studies?**

From the Fig. 4R1 we can see the mean size distribution during the hygroscopic case investigated in this manuscript. The number size distribution has two main peaks at 35 and 115 nm.

**Did the APS see particles above 2-3 µm, which is mentioned to be the upper limit of the Lidar?**

From the mean size distribution (Figure 4 R1) during the hygroscopic case it can be observed that most of the aerosol is in the fine mode (< 1 µm). The upper limit of the lidar is not 2-3 µm, in the manuscript we stated that due to the lidar's wavelengths the instrument is more efficient at those wavelengths. Therefore, to avoid confusion, the following paragraph has been rephrased,

**p14, line 7 - 9** "At SNS station, according to the mean size distribution calculated with SMPS during the hygroscopic growth case two main peaks were detected at 35 and 115 nm (from 20:00 to 21:00 UTC), therefore, it can be observed that most of the aerosol is in the fine mode (< 1 µm), but the lidar's wavelengths used are more efficient at those wavelengths."

**Can dust particles be excluded during the case studies (which could lead to a lower backscatter coefficient but cannot be measured by either ACSM or Aethalometer)?**

During this campaign, the chemical composition was measured with the ACSM and the Aethalometer. This set-up does not allow us to determine the mass concentration of dust particles neither to detect their presence. The air mass trajectories arriving at SNS during 16 June 2016 were coming from the North Atlantic Ocean and the atmospheric situation during this day and the previous days was characterized by clean conditions. Considering also additional data gathered during the campaign at SNS we can conclude that dust particles were not present during the case investigated here. In this sense, the scattering Angstrom Exponent (SAE) can be used as an indicator of the predominance of fine (SAE around 2) or coarse particles (SAE < 1). From the scattering coefficients measured with a nephelometer (TSI 3563) at SNS station, the calculated SAE was > 1.2 during the hygroscopic case under study. In addition, AOD-derived Angstrom Exponent from AERONET Level 2.0 were above 1.4 during the whole day (16 June 2016). The fine mode fraction retrieved by AERONET was above 0.83 evidencing that coarse particles did not predominate.

**It is mentioned that it is difficult to assess the uncertainties in the in-situ data, but can any upper or lower limits be estimated?**

The uncertainty in the f (RH) calculation is a function of the individual measurement uncertainties in the particle number size distribution and in the chemical composition. Furthermore, the different assumptions made to calculate the backscattering coefficients within the Mie model contribute to this uncertainty as well. These assumptions are:

1) Since no experimental measurements of the growth factor, g (RH), were available during the campaign, the g (RH) was calculated based on g(RH) from the literature for the individual chemical components measured with the ACSM+Aethalometer.

2) The change in the size distribution with changing RH is calculated based on a constant and size-independent g (RH).

3) The change in refractive index with changing RH is calculated from a volume-weighting of the aerosol dry refractive index and the refractive index of water, using the mean g (RH) (Hale and Querry, 1973).

This procedure has been extensively used and it has been proven to provide successful results in closure studies (e.g. Fierz-Schmidhauser et al., 2010ACP; Zieger et al., 2013). Concerning the uncertainties in the retrieval, Adam et al. (2012) calculated f (RH) using Mie theory and performed a sensitivity study concluding that errors in the backscattering enhancement factor can vary from 10 up to 30% as the RH changes from <50% to 90%. This information has been included in the revised manuscript.

The following information is now included in the manuscript:

p9, line 11 - 17 "Estimations of $f_\beta$ $(\lambda, RH)$ uncertainty are very scarce because of their high complexity. Some studies (e. g. Adam et al., 2012; Zieger et al., 2013) provided estimations based on sensitivity analysis using Mie model calculations, reporting errors around 20% $f_\sigma$ $(\lambda, RH)$. where σ is the scattering coefficient. Titos et al. (2016) reported uncertainty estimations based on Monte-Carlo techniques, concluding that the more hygroscopic the aerosol, the higher is the uncertainty in $f_\sigma$ $(\lambda, RH)$ especially at high RH (RH > 80%). For moderate-hygroscopic aerosol, it was established a lower limit for the uncertainty in $f_\sigma$ $(\lambda, RH)$ of around 30-40% using nephelometry techniques."

**Also, the size at which the hygroscopic growth factors listed in Table 1 were measured should be mentioned and a short comment on the mean/median size measured during the cases in this study should be added.**

We have included the diameters in the caption of Table 1 and a short comment on the mean size distribution, as suggested by the reviewer.

**p14, line 6 – 9** "At SNS station, according to the mean size distribution calculated with SMPS during the hygroscopic growth case two main peaks were detected at 35 and 115 nm (from 20:00 to 21:00 UTC), therefore, it can be observed that most of the aerosols are in fine mode ($< 1$ µm), but the lidar's wavelengths used are more efficient at those wavelengths."

**Additionally, the introduction would profit from some information on the used method to retrieve in-situ hygroscopicity values (more comments on this are presented later).**

We agree with the referee#2. A detail response is given in the next comment.

**p2, line 24** "Several studies have been carried out over the past years in order to evaluate how water uptake affects aerosol properties. One parameter used to quantify these changes is the so-called aerosol hygroscopic enhancement factor, f ($\lambda$,RH), defined as the ratio between aerosol optical/microphysical properties at wet atmospheric conditions and the corresponding reference value at dry conditions (Hänel 1976; Ferrare et al. 1998; Feingold et al., 2003; Veselovskii et al., 2009; Granados-Muñoz et al., 2015; Titos et al., 2014, 2016, and references therein). Most of the previous studies investigating aerosol hygroscopicity are based on in-situ measurements. One of the most commonly used in-situ instruments for measuring aerosol hygroscopicity is the Humidified Tandem Differential Mobility Analyzer (HTDMA) (e.g. Swietlicki et al., 2008) that measures the hygroscopic growth factor g(RH) that quantifies the change in particle diameter due to water uptake. Humidified tandem nephelometers have been extensively used as well to quantify the effect of the hygroscopic growth in the aerosol optical properties, namely scattering, backscattering and extinction coefficients (e.g., Pilat and Charlson 1966; Titos et al., 2016). There are other in-situ instruments such as the white-light humidified optical particle spectrometer (WHOPS) (Rosatti et al., 2015) or the Differential Aerosol Sizing and Hygroscopicity Spectrometer Probe (DASH-SP), (Sorooshian et al., 2008) that have been used to determine the impact of enhance RH on the aerosol properties from airborne platforms. The effect of RH on the aerosol optical properties can be also determined with Mie model calculations (e.g. Adams et al.,

2012; Fierz-Schmidhauser et al., 2010; Zieger et al., 2013) using the measured size distribution and chemical composition as inputs. For this calculation, the g (RH) is also needed as input. This factor can be determined experimentally (using HTDMA measurements for example) or it can be inferred from the individual growth factors of the different chemical compounds. The assumption of some aerosol properties such as the refractive index or the growth factor based on the chemical composition is the main drawback of this method. Generally speaking, one important limitation of most in-situ techniques is that they modify the ambient conditions and are also subject of particles losses in the sampling lines, therefore altering the real atmospheric aerosol properties."

**General comments:**

**P2, line 30: Please make the section on commonly used hygroscopicity measurements clearer. The HTDMA has only been employed on the ground, whereas there are other instruments for airborne measurements (DASH-SP, WHOPS, AMS+Aethalometer...). Also add some information on the method used in this article and some pros and cons as stated for the HTDMA.**

According to the referee#2's suggestion we have clarified this. We have explicitly mentioned the different instruments used to investigate aerosol hygroscopicity from the ground and from airborne platforms but we have not extended the discussion much further since a review of techniques is not the focus of this manuscript. These lines have been replaced in the manuscript by. For detailed information see the comment above.

**P6, line 6: Please add if any corrections were applied to the Aethalometer data and which MAC value was used to convert the absorption coefficient to the eBC concentrations (and reference).**

No corrections were applied to the Aethalometer AE33 data and the MAC used was 7.77 $m^2g^{-1}$, as recommended by the manufacturer for this wavelength (Drinovec et al., 2014). This information has been added to the instrumentation section.

**p7, line 13 – 16** "Equivalent black carbon, eBC, mass concentration was obtained from measurements of the Aethalometer AE33 at 880 nm. A mass absorption cross section of 7.77 m$^2$g$^{-1}$ was used to convert the absorption coefficients at 880 nm in eBC mass concentrations (Drinovec et al., 2015)"

**Figures: Please use the descriptions of a) b) c) . . . in the text to refer to certain parts of the figures as this facilitates to follow the discussion. Also add legends to Figure 1 and mention what the γ values given in the legends are (the text states only "sold lines").**

We agree with the referee#2. These changes have been performed in the manuscript.

**Title of section 4.2.2: "measured and modelled fβλ(RH)" is a little misleading as no modelling (except HYSPLIT) was performed. Maybe rather use "measured fβλ(RH) and calculated using in-situ data and Mie theory" or something similar**

Following the referee#2's suggestion, we have decided to modify the title as it is follow:

**P13, line 29** "f$_\beta^\lambda(RH)$ measured and retrieved by combining in-situ data and Mie theory"

**Figure 5: The "measured" data exhibits some kind of jump at RH=95%. Can you comment on this and why, possibly, it is not seen in the Mie calculations with the in-situ data?**

The jump seen on figure 5 on measured data, could be associated with the uncertainties on the *RH* calculation itself mostly above *RH*>95%, because of the combination of MWR temperature and calibrated WVMR signal. On this hand, the *RH* profile could have some data tendency to keeping constant meanwhile backscatter coefficient could remain increasing above *RH*=95 %, which may cause to the final calculation an increase of the uncertainty of f (RH).

**Specific comments:**

**P3, line 26: change "on one hand" to "on the one hand"**

Done

**P4, line 5 and following: please rephrase the second part of the sentence with the case of RH>60%**

We have rephrased it in the manuscript:

**p5, line 6 - 8** "Navas-Guzmán et al. (2014) analyzed one year of measurements of $RH$ profiles at Granada showing that RH values are low (below 60%) in the 75% of the cases studied within 1.0 and 2.0 km a.s.l. This study also showed that most of the cases with $RH$ above 60% are found in spring and winter seasons."

**P4, line 23: please specify "incomplete overlap"**

We have included the following sentence in the manuscript:

**p5, line 29-30** "Atmospheric information retrieved from lower regions is limited by the full overlap height, which is reached above 1.3 km a.s.l due to the system configuration (Guerrero-Rascado et al., 2010; Navas-Guzmán et al., 2011)"

**P6, line 6: change "werer" to "were"**

Done

**P7, line 18: which GDAS resolution (degrees) was used?**

We have included in the manuscript:

**p8, line 22 - 23** "GDAS meteorological data used have a spatial resolution of 0.5°x0.5° available since 2010 with daily files every three hours on the native GFS hybrid sigma coordinate system."

**P13, line 6: missing space between "similar" and "γ"**

Done

**P13, line 8: change to "in one of their case studies"**

Done

**P13, line 24: what does the 4% refer to exactly?**

It means that taking as the reference $f_\beta^{355}$ and $f_\beta^{532}$ from Mie calculations and making the relative error calculation with measured values for each wavelength, this relative error remains below 5 %. The relative error for $f_\beta^{355}$ =2.7 % and relative error for $f_\beta^{532}$ =4.3 %.

**P13, line 29: change to "associated with. . ."**

Done

**P13, line 31: change to "reported in.."**

Done

**P14, line 22: use past tense i.e. "were" instead of "are"**

Done

**P14, line 28: change "it is concluded" to "we concluded"**

Done

**P15, line 5: change "gamma parameter" to "γ parameter"**

Done

**P15, line 6: please add a more precise description of what the 13% and 10% describe**

We have commit an error in the percentages, to be precise, we found good agreement on $\gamma$ parameters measured ($\gamma^{532} = 0.48 \pm 0.01$ and $\gamma^{355} = 0.40 \pm 0.01$) respect to Mie calculated ones ($\gamma^{532} = 0.53 \pm 0.02$ and $\gamma^{355} = 0.45 \pm 0.02$), with relative differences between measured and calculated of 9 % at 532 nm and 11 % at 355 nm, taking the calculated values as reference.

**P15, line 7: change "those" to "these"**

Done

**P15, line 7-8: explain what "favorable" means; change "no-advection" to "in absence of advected air masses" or something similar**

Favorable refers to the good atmospheric conditions needed to evaluate these phenomena like vertical homogeneity (good mixed layers), same origin of the aerosols within the analyzed layer, low horizontal velocity and the absence of advected air masses into the evaluated column.

**P15, line 11: change to "making it possible"**

Done

**P15, line 12: change "those" to "these" or "such"**

Done

**Hygroscopic growth study in the framework of EARLINET during the SLOPE I campaign: synergy of remote sensing and in-situ instrumentation**

Andrés E. Bedoya-Velásquez [1,2,3], Francisco Navas-Guzmán [4], María J. Granados-Muñoz [5], Gloria Titos [1,6], Roberto Román[1,2,11], Juan A. Casquero-Vera [1,2], Pablo Ortiz-Amezcua [1,2], Jose A. Benavent-Oltra [1,2], Gregori de Arruda Moreira[1,2,7], Elena Montilla-Rosero [8], Carlos D. Hoyos [9], Begoña Artiñano [10], Esther Coz [10], Francisco José Olmo-Reyes[1,2], Lucas Alados-Arboledas [1,2] and Juan Luis Guerrero- Rascado [1,2].

[1]Andalusian Institute for Earth System Research (IISTA-CEAMA), University of Granada, Autonomous Government of Andalusia. 18006, Granada, Spain.
[2]Departament of Applied Physics, University of Granada. Granada, Spain.
[3]Sciences Faculty, Department of Physics, Universidad Nacional de Colombia. Medellín, Colombia.
[4]Federal Office of Meteorology and Climatology MeteoSwiss, Payerne, Switzerland.
[5]Remote Sensing Laboratory / CommSensLab, Universitat Politècnica de Catalunya, Barcelona, 08034, Spain.
[6]Institute of Environmental Assessment and Water Research (IDAEA), CSIC, Barcelona, 08034, Spain.
[7]Institute of Research and Nuclear Energy, IPEN. São Paulo, Brazil.
[8]Physical Sciences Department, School of Science, EAFIT University, Medellín, Colombia.
[9]Minas Faculty, Department of Geosciences and Environment, Universidad Nacional de Colombia.Medellín, Colombia.
[10]CIEMAT, Environment Department, Associated Unit to CSIC on Atmospheric Pollution, Avenida Complutense 40, Madrid, Spain.
[11]Grupo de Óptica Atmosférica (GOA), Universidad de Valladolid.  Paseo Belén, 7, 47011, Valladolid, Spain.

*Correspondence to*: Andrés Esteban Bedoya Velásquez (aebedoyav@correo.ugr.es)

**Abstract**

This study focuses on the analysis of aerosol hygroscopic growth during Sierra Nevada Lidar AerOsol Profiling Experiment (SLOPE I) campaign by using the synergy of active and passive remote sensors at ACTRIS Granada station and in-situ instrumentation at a mountain station (Sierra Nevada, SNS). To this end, a methodology based on simultaneous measurements of aerosol profiles from an EARLINET multi-wavelength Raman lidar (RL) and relative humidity (RH) profiles obtained from a multi-instrumental approach is used. This approach is based on the combination of calibrated water vapour mixing ratio (r) profiles from RL and continuous temperature profiles from a microwave radiometer (MWR) for obtaining RH profiles with a reasonable vertical and temporal resolution. This methodology is validated against the traditional one that uses RH from co-located radiosounding (RS) measurements, obtaining differences in hygroscopic growth parameter ($\gamma$) lower than 5% between the methodology based on RS and the one presented here. Additionally, during SLOPE I campaign the remote sensing methodology used for aerosol hygroscopic growth studies has been checked against Mie calculations of aerosol hygroscopic growth using in-situ measurements of particle number size distribution and submicron chemical composition measured at SNS.

none
The hygroscopic case observed during SLOPE I showed an increase in particle backscatter coefficient at 355 and 532 nm with relative humidity (RH ranged between 78 to 98%), but also a decrease in backscatter-related Ångström exponent (AE) and particle linear depolarization ratio (PLDR) indicating that the particles became larger and more spherical due to hygroscopic processes. Vertical and horizontal wind analysis is performed by means of a co-located Doppler lidar system, in order to

5    evaluate the horizontal and vertical dynamics of the air masses. Finally, the Hänel parameterization is applied to experimental data for both stations and we found good agreement on $\gamma$ measured with remote sensing ($\gamma^{532} = 0.48 \pm 0.01$ and $\gamma^{355} = 0.40 \pm 0.01$) respect to the values calculated using Mie theory ($\gamma^{532} = 0.53 \pm 0.02$ and $\gamma^{355} = 0.45 \pm 0.02$), with relative differences between measurements and simulations lower than 9 % at 532 nm and 11 % at 355 nm.

10   KEYWORDS: ACTRIS, Aerosol hygroscopic growth, Doppler lidar, EARLINET, lidar, microwave radiometry, remote sensing.

**1 Introduction**

Atmospheric aerosol particles play a crucial role in the Earth´s climate, principally by means of the radiative effect due to

15   aerosol-radiation and aerosol-cloud interactions, affecting the Earth-atmosphere energy balance and, hence, the Earth's climate. Furthermore, aerosol might also modify optical and microphysical cloud properties, such as albedo and cloud droplet size distribution that influences cloud lifetime, since the particles could act as cloud condensation nuclei (CCN) and ice nuclei (IN) (Twomey, 1977; Albrecht, 1989; IPCC, 2013).

20   Water vapor plays a major role in the aerosol-radiation interaction due to the ability of some atmospheric aerosol particles to take up water from the environment. In this sense, hygroscopic growth is the process by which aerosol particles uptake water and increase their size under high relative humidity (RH) conditions (Hänel, 1976). Consequently, this process is also related to changes in the optical and microphysical properties of the aerosol particles and, hence, it becomes a crucial factor that modifies the role of aerosols in atmospheric processes and radiative forcing.

Several studies have been carried out over the past years in order to evaluate how water uptake affects aerosol properties. One parameter used to quantify these changes is the so-called aerosol hygroscopic enhancement factor: f (λ, RH), where λ is the wavelength, defined as the ratio between aerosol optical/microphysical properties at wet atmospheric conditions and the corresponding reference value at dry conditions (Hänel 1976; Ferrare et al. 1998; Feingold et al., 2003; Veselovskii et al.,

30   2009; Granados-Muñoz et al., 2015; Titos et al., 2014, 2016, and references therein). Most of the previous studies investigating aerosol hygroscopicity are based on in-situ measurements. One of the most commonly used in-situ instruments for measuring aerosol hygroscopicity is the Humidified Tandem Differential Mobility Analyzer (HTDMA) (e.g. Swietlicki et al., 2008) that

measures the hygroscopic growth factor, g(RH), that quantifies the change in particle diameter due to water uptake. Humidified tandem nephelometers have been extensively used as well to quantify the effect of the hygroscopic growth in the aerosol optical properties like scattering, backscattering and extinction coefficients (e.g., Pilat and Charlson 1966; Titos et al., 2016). There are other in-situ instruments such as the white-light humidified optical particle spectrometer (WHOPS) (Rosatti et al.,

5   2015) or the Differential Aerosol Sizing and Hygroscopicity Spectrometer Probe (DASH-SP) (Sorooshian et al., 2008) that have been used to determine the impact of enhanced RH on the aerosol properties from airborne platforms.

The effect of RH on the aerosol optical properties can be also determined with Mie model calculations (e.g. Adams et al., 2012; Fierz-Schmidhauser et al., 2010; Zieger et al., 2013) using the measured size distribution and chemical composition as inputs. For this calculation, information on g (RH) is needed a priori. This factor can be determined experimentally (using

10   HTDMA measurements for example) or it can be inferred from the individual growth factors of the different chemical compounds. The assumption of some aerosol properties such as the refractive index or the growth factor based on the chemical composition is the main drawback of this method.

In general terms, most in-situ techniques are limited by the fact that they modify the ambient conditions and are also subject

15   to particles losses in the sampling lines, therefore altering the real atmospheric aerosol properties. Remote sensing systems such as lidars have also been used in the last decades for aerosol hygroscopic growth studies performed with co-located radiosounding (RS) measurements (e. g. Ferrare et al., 1998; Feingold et al., 2003; Veselovskii et al., 2009; Granados-Muñoz et al., 2015; Fernández et al., 2015; Lv et al., 2017). These systems have shown to be robust with high vertical and temporal resolution that allow for studying the aerosol hygroscopic growth under unmodified ambient conditions. Recent studies

20   presented by Zieger et at. (2011) and Rosati et al. (2016) show good agreement between in-situ and RL extinction coefficients after taking into account the RH effect on the in-situ measured extinction coefficient. Also, it is possible to use aerosol extinction coefficient to compare with in-situ airborne measures and elastic lidar to study hygroscopic growth in unmodified ambient conditions. In addition, good results were obtained by using Automatic Lidar and Ceilometers (ALC) to investigate hygroscopic growth and fog formation, mostly for fog events forecasting purposes (Haeffelin et al., 2016).

Up to now, most hygroscopic growth studies using lidar systems combine lidar measurements with RH data from RS (Granados-Muñoz et al., 2015). The main inconveniences are that RS measurements have low temporal sampling and they could be drifted away from the vertical atmosphere probed by the lidar systems. These inconveniences can be easily overcome by combining calibrated water vapor mixing ratio profiles, r (z) from Raman lidar (RL), with temperature profiles from

30   ancillary instrumentation for obtaining collocated RH and aerosol backscatter profiles, using them simultaneously for hygroscopic growth studies (e.g. Whiteman, 2003; Navas-Guzmán et al., 2014; Barrera-Verdejo et al., 2016). Navas-Guzmán et al. (2014) proposed a methodology for retrieving RH profiles by the combination of calibrated r (z) profiles from Raman lidar water vapor channel with temperature profiles obtained from microwave radiometer (MWR) measurements. RH profiles obtained using this multi-instrumental approach and aerosol profiles from the lidar are used in this work to study aerosol

hygroscopic growth. This methodology allows to obtain a larger database of potential hygroscopic cases since some of the limitations associated to RS are overcome. Additionally, water vapour and aerosol measurements are performed with the same system and, thus, the same air volume is probed, avoiding the possible radiosonde drift and temporal sampling mismatch.

5   The main goal of this study is to apply the methodology proposed by Navas-Guzmán et al. (2014), based on the application of the synergy between RL and MWR, for aerosol hygroscopic growth studies. First, this methodology for hygroscopic growth studies is compared with the approach presented in Granados-Muñoz et al. (2015) that uses RS and lidar data. Once the technique is evaluated, an analysis of the aerosol hygroscopic growth case observed during the SLOPE I (Sierra Nevada Lidar AerOsol Profiling Experiment I) campaign is presented. In addition, the results obtained with the remote sensing data are
10  compared with Mie simulations performed using in-situ measurements from a high-mountain station located at 2500 m a.s.l.

This paper is organized as follows. The description of the experimental site and instrumentation is presented in Section 2. The applied methodology is introduced in Section 3. Section 4 presents the results and discussion of the combination of RL and MWR measurements for obtaining RH profiles and the analysis of the aerosol hygroscopic cases based on the remote sensing
15  and in-situ measurements. Finally, conclusions are given in Section 5.

 **2 Experimental site and instrumentation**

**2.1 SLOPE I field campaign**

In summer 2016, the Sierra Nevada Lidar AerOsol Profiling Experiment (SLOPE I) intensive field campaign was carried out in South-Eastern Spain in the framework of the European infrastructure ACTRIS. The goal of this campaign was to perform a closure study by comparing remote sensing and in-situ measurements at different altitudes taking advantage of a unique experimental setup (Román et al., 2018). This setup consisted of several experimental stations located at different altitude
25  levels in the slope of Sierra Nevada, located 20 km away in horizontal distance from the remote sensors at IISTA-CEAMA station (urban station at Granada). In the present study, we only make use of the data from the in-situ instrumentation of the mountain Sierra Nevada station (SNS) located at 2500 m a.s.l, SNS in Fig. 1. Combined active and passive remote sensing measurements using multiple instrumentation at the Andalusian Institute of Earth System Research (IISTA-CEAMA) station and simultaneous in-situ measurements at 2500 m a.s.l in the northern slope of Sierra Nevada were performed from May to
30  September 2016 during this campaign. In addition, 25 RS were launched during this period, 6 of them during night-time in order to perform regular calibration of the Raman lidar water vapour channel.

**2.2 IISTA-CEAMA station**

One of the stations where this study has been carried out is the IISTA-CEAMA, an urban station managed by the University of Granada (UGR) located at Granada, Spain (37.16° N, 3.61° W, 680 m a.s.l.). This region is characterized by its complex terrain surrounded by mountains, mainly affected by Mediterranean continental climate conditions with hot summers and cool winters. Navas-Guzmán et al. (2014) analyzed one year of measurements of RH profiles at Granada showing that this location presents low values of RH (below 60%) in the 75% of the cases studied for altitudes between 1.0 and 2.0 km a.s.l. RH values above 60% are mostly found in spring and winter seasons. Regarding the remote aerosol sources, Granada is predominantly affected by aerosol particles coming from Europe and mineral dust particles from the African continent (Lyamani et al., 2006a, b; Guerrero-Rascado et al., 2008a, 2009; Córdoba-Jabonero et al., 2011; Titos et al., 2012; Navas-Guzmán et al., 2013; Valenzuela et al., 2014; Benavent-Oltra et al., 2017; Cazorla et al., 2017). Main local sources are road traffic, domestic-heating (during wintertime) and biomass burning (Titos et al., 2017). Transported smoke principally from North America, North Africa and the Iberian Peninsula can also affect the study area (Alados-Arboledas et al., 2011; Navas-Guzmán et al., 2013; Preißler et al., 2013; Pereira et al., 2014; Ortiz-Amezcua et al., 2017). Moreover, the probability of marine particles reaching the city is low despite the short distance to the coast (about 50 km away) due to the orography of the region, with mountains blocking the path from the sea to the city. Additionally, Titos et al. (2014) showed that the contribution of marine aerosols to PM10 mass concentration at IISTA-CEAMA station is almost negligible (<3%).

The main instrument used in this study and located at IISTA-CEAMA station is the multi-wavelength Raman lidar (RL) MULHACEN (Raymetrics S.A., Greece). MULHACEN is included in EARLINET (European Aerosol Lidar NETwork) (Pappalardo, et al., 2014), now operating in the framework of ACTRIS-2 (Aerosols, Clouds and Trace gases Research Infrastructure). It emits laser pulses at 355 and 532 nm (parallel and perpendicular polarization channels) and 1064 nm, and it receives backscattered photons at 355, 532 and 1064 nm in analog and photon counting modes. It also collects Raman backscattered photons at 607 and 387 nm from molecular nitrogen ($N_2$) and at 408 nm from 
[revised manuscript text omitted]

In this work, we have also checked the RH calculation (see Eq.1) for the case of 22$^{nd}$ June 2013 by using temperature profiles from MWR and GDAS modelled data which were compared to RS RH profiles. This comparison allows us to investigate the feasibility of the use of GDAS temperature information to compute the RH profiles in combination with RL profiles, in order
30   to increase the database for hygroscopicity studies. However, the results present larger bias when they are compared with the RS HR profiles, up to 20 % for RH$_{LIDAR+GDAS}$ in almost the whole profile instead of the10% for the RH$_{LIDAR+MWR}$ (Fig. 2). Thus, the use of GDAS data seems not to be appropriate in this study, mainly because of two reasons: (i) the complex terrain

where the measurement station is located, surrounded by mountains of high elevation (up to more than 3000 m a.s.l. in a very short horizontal distance of few tenths of kilometres) that makes more difficult for models to provide accurate thermodynamics profiles for this location; (ii) GDAS profiles have a lower temporal resolution (3 h) than the MWR, which gives temperature profiles each 2 min.

Figure 3 shows, from left to right, the RH profiles obtained from both the RS (black line) and the synergy RL+MWR (red line), the bias between both profiles (RH $_{RS}$ − RH $_{RL + MWR}$), $\beta$ $_{532nm}$ profiles retrieved from the lidar system and $f_\beta$ (RH). The upper panels correspond to case I on 22$^{nd}$ July 2011 and the bottom panels to case II on 22$^{nd}$ July 2013. Horizontal dashed lines mark the region of interest analysed for each case, ranging from 1.3 to 2.3 km a.s.l. for case I and 1.3 to 2.7 km a.s.l. for case II.

RH profiles (Fig. 3 a and e, red line) calculated by the combination between RL calibrated r (z) profile and MWR temperature profiles were obtained following the methodology presented in section 3.1 by using Eq. 3 (Navas-Guzmán et al., 2014). Good agreement is observed, with biases (Fig. 3 b and f) lower than 10% within the analysed region. The differences obtained in the RH profiles might be associated to the discrepancies between the temperature profiles from MWR and RS, due to the lower

15  vertical resolution of the MWR. Additionally, discrepancies are also expected because of the radiosonde drift and the different temporal sampling (the lidar data correspond to a 30-min average, whereas the RS provides instantaneous values that build the profile in the region of interest in less than 5 minutes).

The discrepancies between the two RH profiles are especially relevant in the lower part of the analyzed data since differences

20  of RH in this region lead to variations in RH$_{ref}$. For case I, RH$_{ref}$ = 60% for RS and RH$_{ref}$ = 68% for the RL+MWR combination, whereas for case II, RH$_{ref}$ = 40% for RS and RH$_{ref}$ = 50% for RL + MWR methodology. Additionally, the RH discrepancies in the upper region of the profiles (from 2.1 to 2.3 km a.s.l. for case I and from 2.6 to 2.7 km a.s.l. for case II), which can reach up to 5%, are also relevant since they are associated to the maximum values of RH and may modify the data tendency on Hänel's parameterization, leading to variations in γ (λ) depending on the methodology used for the retrieval

25  of RH. Despite these discrepancies, the differences between γ (λ) parameters obtained from both methodologies are low (Table 2). On case I, γ (λ) = 0.59 ± 0.05 obtained from RL + MWR is larger than that obtained from RS (γ = 0.56 ± 0.01). While on case II the γ obtained with RH from RS (γ = 0.99 ± 0.01) is larger than the one from RL+MWR (γ = 0.95 ± 0.02). We have to keep in mind that uncertainties reported on γ are obtained by the polynomial fitting and they do not include the propagation error result. The relative differences on both cases are below 5%, which is relatively good compared to the

30  expected uncertainties reported in Titos et al., (2016) and considering the differences between the two methodologies.

The obtained values of $f_\beta$ (85%) using both methodologies are presented in Table 2. For case I, $f_\beta$ (85%) = 1.50 for RS and $f_\beta$ (85%) = 1.46 for RL+MWR, with a relative difference below 3%. For case II, $f_\beta$ (85%) = 2.6 for RS and $f_\beta$ (85%) =

2.3 for RL+MWR showing a relative difference of 11%. Even though the relative difference is larger for case II, for both cases the discrepancies lie within the uncertainty associated to the calculation of $f_\beta$ (85%) which is around 20% according to Titos et al. (2016). Thus, the RL+MWR methodology presented by Navas-Guzmán et al. (2014) to obtain RH profiles in a continuous time base is a promising technique for hygroscopic growth studies. This methodology will allow for expanding the RH profile

5  database and it opens new opportunities for the detection of hygroscopic cases during night-time periods.

**4.2. Hygroscopic study during SLOPE I**

**4.2.1 Conditions for hygroscopic growth**

Aerosol hygroscopic growth was observed during SLOPE I campaign in 2016 combining the remote sensing instruments and the RS. Figure 4 shows the time series of range corrected signal (RCS) at 532 nm derived by the EARLINET lidar system at the IISTA-CEAMA station on 16[th] June 2016. The presence of clouds is observed in the late afternoon (~3.0 km a.s.l.) before 19:00 UTC, with clouds vanishing after that during the remaining measurement period. The red lines in Fig. 3 mark the 30

15  min set of profiles (from 20:30 to 21:00 UTC) where an intensification of the RCS is observed at 2.5 km a.s.l, which could be an indication of potential aerosol hygroscopic growth.

Figure 5 shows profiles of r (z), $\theta_v$, RH, $\beta_{par}$ at 355 and 532 nm, backscatter-related Ångström exponent between 355 and 532 nm (AE $_{355-532}$) and PLDR $_{532}$ (particle linear depolarization ratio at 532 nm) obtained on 16[th] June 2016 between 20:30 and

20  21:00 UTC. As we mentioned in section 3.2, for aerosol hygroscopicity analysis it must be ensured that ranges where RH increases correspond to an increase in $\beta_{par}$, which is well seen along the layer between 1.5 and 2.4 km a.s.l. (see Fig. 5). The RH profile was calculated by using the method combining RL+MWR. In this case, the calibration constant for the RL r (z) profile was calculated using the six RSs launched at night-time during this campaign. A calibration constant of $110 \pm 2$ g/kg was obtained as the mean value of the different calibrations.

In order to fulfil all the requirements discussed in sections 3.2 and 3.3 for hygroscopic growth studies, together with the RH and $\beta_{par}$ increase within the layer, atmospheric stability must be ensured through the evaluation of thermodynamic variables such as $\theta_v$ and r (z). Here, r (z) shows relatively low vertical variation within the region of interest (1.5 to 2.4 km a.s.l.), decreasing monotonically with altitude at a rate of $-1.9 \frac{g}{kg \cdot km}$(Fig. 5a) and $\theta_v$ shows a monotonic increase at a rate $\frac{\partial \theta_v}{\partial z} =$

30  $0.03 \frac{°C}{km}$ within the same region.

AE $_{355-532}$ and PLDR $_{532}$ were also retrieved in order to describe the mean size and shape of the aerosol particles. For this case, we observe a decrease in both parameters in the region of interest. A decrease in AE $_{355-532\,nm}$ ($\sim$0.4 km$^{-1}$) means an increase in the predominance of larger particles and a decrease of the PLDR $_{532\,nm}$ ($\sim$0.13 km$^{-1}$) is related to particles becoming more spherical. This correlation between AE $_{355-532}$ and PLDR has been observed in previous studies associated to hygroscopic growth (Granados-Muñoz et al., 2015; Haarig et al., 2017).

In order to determine the origin of the aerosol particles over the analysed layer, we present a horizontal wind speed and direction and vertical wind analysis from Doppler lidar data. The 10 min resolved horizontal wind direction time series (Fig. 6b) indicate that from 18:00 to 21:00 UTC the wind over IISTA-CEAMA station mainly came from the North-West, within the region of interest (1.5 to 2.4 km a.s.l.) with relative low horizontal wind velocity (up to 6 m/s) (Fig. 6a), which means that aerosol particles were being transported from the same direction, likely coming from the same source, at relative low horizontal velocity.

A turbulence analysis was also performed to reinforce the fact that vertical fluxes within the aerosol column are associated to increases of RCS observed in Fig. 4. The aerosol RCS increases in a region where RH increases as we see in Fig. 5, thus we associate this increases in RCS with water uptake by aerosols inside this atmospheric column. The vertical wind velocity can be statistically studied to obtain the higher moments of the velocity distribution (O'Connor et al., 2010, Moreira et al., 2018a). This statistical analysis is deeply developed for turbulence studies. Here the third moment of the frequency distribution (skewness) (Fig. 6c) represents the direction of the convection (positive skewness is associated to predominance of upward wind velocity whereas negative skewness means predominance of downward wind) in the region of interest. Supporting this analysis, the black stars represent the calculation of the atmospheric boundary layer height (PBLH, Fig. 6c) obtained from the MWR data by using the combination of parcel and gradient methods in convective and stable atmospheric conditions (Holzworth, 1964; Moreira et al., 2018b). In this case, close to 21:00 UTC (Fig. 6c), the particles tend to ascend into the column, as indicated by positive values reached in the skewness linked with highly convective movement. The PBLH reach its maximum at 15:00 UTC (2.5 km a.s.l.) but after 16:00 UTC the weakening of convection tends to decrease the ABLH, keeping the ABLH around 2 km a.s.l. until 21:00 UTC. All this wind information might be interpreted as transported particles coming from the same direction at relative low horizontal velocities, suggesting that aerosol source is not changing and new aerosol particles are not being advected into the studied layer. The turbulence analysis allows us to support that vertical wind movement within the layer of interest drives to well mixing processes during the analysed time interval.

The 6-day backward trajectories were calculated at three different heights (0.9, 1.5 and 1.9 km a.g.l), which were selected within the region of interest in order to guarantee the height-independency of the air masses pathway. The three air masses came from North America, crossing the Atlantic Ocean, reaching the continental platform through Portugal and then advected

to Granada reaching the station at 21:00 UTC (not shown here). This information supports the horizontal wind analysis performed before.

**4.2.2 $f_\beta^\lambda$ (RH) measured and retrieved by combining in-situ data and Mie theory**

The humidogram presented in Fig. 6 shows the measured $f_\beta^\lambda$ (RH) at 355 and 532 nm as a function of RH between 1.5 and 2.4 km a.s.l, retrieved by using the lidar data. The calculated $f_\beta^\lambda$(RH) was obtained by using the measured chemical composition and size distribution at SNS station (2.5 km a.s.l.) as inputs to the Mie model (see Table 1 and Fig. 7). The humidogram exhibits a monotonic positive increase at both wavelengths, for RH between 78 and 98%. The RH $_{ref}$ = 78 % was selected as
10 the lowest RH value into the evaluated column, and this same RH was used as reference for the Mie calculation in order to make both calculations comparable.

During the hygroscopic growth event at SNS station, the mean aerosol particle number size distribution shows two main peaks at around 35 and 115 nm, with most of the aerosol in the fine mode (< 1 µm). The sub-micron mass concentration measured
15 with ACSM indicates high concentration of organic particles during daytime (from 12:00 to 17:00 UTC), with values around 7 µg/m$^3$ at 15:00 UTC. OA concentrations decreased slowly to values around 3.0 µg/m$^3$ at 00:00 UTC. In particular, during the hygroscopic growth case under study (from 20:00 to 21:00 UTC) the aerosol composition was mainly made up of organic particles (62%) followed by sulphate (24%), nitrate (10%), ammonia (2%) and black carbon (2%). Thus, the predominant aerosol studied during the event is a combination of smoke and urban polluted aerosol. This assumption about the aerosol type
20 is supported by the relatively high sulphate concentration observed at SNS and the results discussed in section 4.2.1 (lidar properties and backward trajectories analyses). This chemical composition with high predominance of organic particles is consistent with the γ values obtained with the RL+MWR method. Fernández et al. (2015) reported a similar γ $^{532}$ value of 0.59 in Cabauw (Netherlands) associated with high concentration of organic particles while they observed a significantly larger γ $^{532}$ of 0.88 associated with marine particles. Lower values are reported by Lv et al. (2017) in one of their case studies
25 (γ $^{532}$ = 0.24 and γ $^{355}$ = 0.12) in Xinzhou (China) associated with the presence of dust particles. Although the behaviour of the backscatter coefficient at enhanced RH is expected to differ from the scattering coefficient, a qualitative comparison can be performed due to the scarcity of backscatter-related γ values in the literature. For example, using in-situ techniques, Zieger et al. (2015) reported a low scattering enhancement of boreal aerosol in Hyytiälä (Finland) (γ $^{525}$ = 0.25) related to the high contribution of organic aerosols at this site that contribute to decrease the hygroscopic enhancement. At Cape Cod (USA),
30 Titos et al. (2014) reported significantly lower γ values for polluted aerosols (γ $^{550}$ = 0.4 ± 0.1) compared with marine aerosols (γ $^{550}$ = 0.7 ± 0.1).

Calculated and measured values of $f_\beta^\lambda$ (RH) are compared in Table 3 and Figure 7. In general, there is a good agreement between measured and calculated hygroscopicity parameters. For both wavelengths, slightly higher values are predicted by the model compared with the measurements, especially at RH > 90% where the differences are higher than at RH < 90%. The values retrieved from the RL measurements are $f_\beta^{355}$ (85%) = 1.07 ± 0.03 and $f_\beta^{532}$ (85%) = 1.20 ± 0.03 and with Mie theory are $f_\beta^{355}$ (85%) = 1.10 ± 0.01 and $f_\beta^{532}$ (85%) = 1.15 ± 0.01. The good agreement found in this analysis is confirmed by the low relative differences observed (lower than 4 %). The hygroscopic growth parameter (γ) shows also good agreement between the measured ($\gamma^{532}$ = 0.48 ± 0.01 and $\gamma^{355}$ = 0.40 ± 0.01) and the calculated ones using Mie theory ($\gamma^{532}$ = 0.53 ± 0.02 and $\gamma^{355}$ = 0.45 ± 0.02), with relative differences of 9 % at 532 nm and 11 % at 355 nm. The good agreement between the measured and theoretical backscatter enhancement factor evidences the robustness of the proposed method for hygroscopic studies in a systematic manner.

The principal sources of error in the comparison between calculated and measured data are associated with the method for the retrieval of RH profiles, as well as the errors associated with theoretical Mie calculation mainly by the assumption of g (RH) based on the chemical composition. Finally, the horizontal distance between stations could also lead to differences in the comparison. The uncertainties affecting our study are the result of the contributions of the particle backscatter uncertainties and experimental uncertainties associated to determination of the backscatter enhancement factor, thus further studies should center their efforts on this research field to constraint the range of uncertainty.

In addition, the multi-wavelength results lead us to see a clear spectral dependence on γ (λ). The efficiency due to changes in $f_\beta^\lambda$ (RH) associated to $\beta_{par}$ is stronger at 532 nm than at 355 nm, finding that $f_\beta^{532}$ (85%) = 1.20 > $f_\beta^{355}$ (85%) = 1.07. This is also seen on the gamma parameter ($\gamma^{532}$ = 0.48 ± 0.01 > $\gamma^{355}$ = 0.40 ± 0.01, with correlations of 0.84 and 0.65, respectively). This spectral dependency has also been reported in Kotchenruther et al. (1999) for in-situ measurements at 450, 550 and 700 nm, obtaining increasing enhancement factors with wavelength, and in Zieger et al. (2013), where the same behaviour is observed for marine aerosols. As it is reported in Haarig et al. (2017) the enhancement factor dependency with wavelength suggests that larger wavelengths have an enhancement factor larger than short ones which in fact was also evidenced on this work.

**5. Conclusions**

The methodology proposed for calculating RH profiles by combining calibrated r (z) from RL and temperature profiles from MWR has been used in this work to study aerosol hygroscopicity. With this method, a way to retrieve RH profiles without the necessity of co-located RS is presented at IISTA-CEAMA station. In order to validate this methodology, hygroscopic growth

cases which use RS data were selected. The relative differences on the $f_\beta^\lambda$ (RH) obtained using the RH profiles from the RS and from the combination of RL and MWR measurements were calculated, finding relative differences below 11 % on $f_\beta$ (85%). The relative differences on γ were below 5 %, supporting the fact that this methodology is valid for aerosol hygroscopicity studies.

Aerosol hygroscopic growth observed during SLOPE I field campaign (16[th] June 2016, 20:30 to 21:00 UTC) was studied by means of particle backscatter coefficient retrieved from the EARLINET multi-wavelength RL, backscatter-related-Ångström exponent ($AE_{355-532}$) and particle linear depolarization ratio ($PLDR_{532}$) as optical properties and the combined RL+MWR RH profiles. Stability analysis confirmed good mixing conditions in the atmospheric layer studied. In addition, Doppler wind lidar data analysis allowed us to evaluate the vertical profiles of horizontal wind velocity and direction. Thus, we concluded that particles came mainly from the North-West region of Granada at low velocities. Furthermore, the skewness analysis let us infer that particles presented an upward movement during the 30 min evaluated period within the column of interest. These results were confirmed by ABLH calculations from MWR data. From the experimental data from RL, values of $f_\beta^{355}$ (85%) = $1.07 \pm 0.03$ and $f_\beta^{532}$ (85%) = $1.20 \pm 0.03$ at $RH_{ref}$ = 78 % were obtained within the evaluated column and also $\gamma^{532}$ = $0.47 \pm 0.01$ ($R^2$=0.84) and $\gamma^{355}$ = $0.40 \pm 0.01$ ($R^2$ = 0.65), which were in agreement with the literature.

For the study case during SLOPE I the results were validated against Mie simulations with experimental data from SNS data obtaining a good agreement between the values retrieved with RL ($f_\beta^{355}$ (85%) = 1.07 and $f_\beta^{532}$ (85%) = 1.20) and Mie theory ($f_\beta^{355}$ (85%) = 1.10 and $f_\beta^{532}$ (85%) = 1.15) reaching relative differences lower than 4% when taking the calculated data as reference. We also found good agreement between the measured hygroscopic growth parameter (γ) ($\gamma^{532}$ = $0.48 \pm$ 0.01 and $\gamma^{355}$ = $0.40 \pm 0.01$) and the calculated one ($\gamma^{532}$ = $0.53 \pm 0.02$ and $\gamma^{355}$ = $0.45 \pm 0.02$), with relative differences up to 9 % at 532 nm and 11 % at 355 nm, taking the calculated data as reference. These results show that under favorable atmospheric conditions (vertical homogeneity, consistent aerosol sources and low horizontal velocity within the analyzed layer) and in the absence of advected air masses into the evaluated column, the hygroscopic behavior of the particles evaluated by remote sensing at IISTA-CEAMA station is in accordance with that evaluated for those particles transported to SNS.

The results obtained here show the potentiality of combining r(z) from RL and temperature from MWR to retrieve RH profiles with high temporal/spatial resolution to analyze aerosol 
[revised manuscript text omitted]

10   Román, R., Benavent-Oltra, J.A., Casquero-Vera, J.A., Lopatin, A. Cazorla, A., Lyamani, H., Denjean, C., Fuertes, D., Pérez-Ramírez, D., Torres, B., Toledano, C., Dubovik, O., Cachorro, V.E., de Frutos, A.M., Olmo, F.J., and Alados-Arboledas, L.: Retrieval of aerosol profiles combining sunphotometer and ceilometer measurements in GRASP. Atmos. Res., 204, 161–167, doi:10.1016/j.atmosres.2018.01.021, 2018.

15   Rosati, B., Herrmann, E., Bucci, S., Fierli, F., Cairo, F., Gysel, M., Tillman, R., Größ, J., Gobbi, G.P., Di Liberto, L., Di Donfrancesco, G., Wiedensolher, A., Weingartner, E., Virtanen, A., Mentel, T.F., and Baltensperger, U.: Studying the vertical aerosol extinction coefficient by comparing in situ airborne data and elastic backscatter lidar, Atmos. Chem. Phys., 16, 4539–4554, doi:10.5194/acp-16-4539-2016, 2016.

[revised manuscript text omitted]

**Figure captions**

**Figure 1.** Topographic profile of Granada and Sierra Nevada area. The yellow star refers to IISTA-CEAMA station and green star refers to SNS in-situ station.

**Figure 2.** RH comparison for 22$^{nd}$ Jul 2013 around 20:00-21_00 UTC. (a) RH profiles retrieved from combination of lidar+MWR (black line), lidar+GDAS (blue line) and RS (red line) and (b) Bias calculation between lidar+MWR (redline), lidar+GDAS (blue line).

**Figure 3.** (a, e) Profiles of *RH* retrieved from *RS* (black line) and by the synergy RL+MWR (red line), (b, f), *RH* bias profiles (cyan line), (c, g) $\beta_{par}$ retrieved by using Klett-Fernald algorithm and lidar ratio of 65 Sr (green line), (d, f) $f_\beta(RH)$ calculated for RS (black dots) and by the synergy RL+MWR (red dots) and the corresponding Hänel parameterizations (solid lines), where red line refers to RL+MWR method (case I: $\gamma = 0.59 \pm 0.05$, case II: $\gamma = 0.95 \pm 0.02$) and black line refers to RS method (case I: $\gamma = 0.56 \pm 0.01$, case II: $\gamma = 0.99 \pm 0.01$) .The top row corresponds to case I (22$^{nd}$ July 2011, 20:30-21:00 UTC) and the bottom row to case II (22$^{nd}$ July 2013, 20:00-20:30 UTC). Horizontal dashed lines indicate the altitude range analysed for each case (1.3 to 2.3 km for case I and 1.3 to 2.7 km for case II). All these profiles were measured at the EARLINET IISTA-CEAMA station.

**Figure 4.** EARLINET IISTA-CEAMA lidar *RCS* time series at 532 nm, 16$^{th}$ June 2016 (17:00 to 00:00 UTC). The sunset estimated for this day was at 21:30 UTC of local time.

**Figure 5.** (a) Water vapour mixing ratio; (b) virtual potential temperature; (c) relative humidity obtained from synergy RL+MWR; (d) particle backscatter coefficient at 355 and 532 nm; (e) backscatter-related Ångström exponent (355-532 nm) and (f) particle linear depolarization ratio. All profiles correspond to a 30 min-average from 20:30 to 21:00 UTC on 16$^{th}$ June 2016 at the EARLINET IISTA-CEAMA station.

**Figure 6.** Time series of (a) horizontal wind velocity, (b) horizontal wind direction and (c) skewness retrieved from Doppler turbulence calculations for 16$^{th}$ June 2016 at 20:30 to 21:00 UTC. The ABLH retrieved from MWR is presented in black stars.

**Figure 7.** Humidograms calculated (a) at 532 nm and (b) at 355 nm, within 1.5 to 2.4 km a.s.l. aerosol layer from the RL+MWR measurements and calculated using Mie theory and measured chemical composition and size distribution at 2.5 km a.s.l. A $RH_{ref} = 78$ % was used for both methods.

**Table 1:**

| | H$_2$0 | OA | NH$_4$NO$_3$ | (NH$_4$)$_2$SO$_4$ | BC |
|---|---|---|---|---|---|
| $m$ (355 nm) | 1.343[a] | 1.458[g] | 1.562[c,f] | 1.56[e] | 1.75 +0.465i[d] |
| $m$ (532 nm) | 1.333[a] | 1.411[g] | 1.556[c] | 1.530[c] | 1.75 + 0.44i[d] |
| $\rho$ | 1 | 1.4[h] | 1.72[i] | 1.77[i] | 1.7[b] |
| $g$ (RH=90%) | -- | 1.05[j] | 1.74[k] | 1.66[k] | 1[l] |

(a) Hale and Querry (1973); (b) Nessler et al. (2005); (c) Fierz-Schmidhauser et al. (2010); (d) Hess et al. (1998)

(e) Ma and Thompson (2012); (f) Linear interpolation to 355 nm (Kou et al., 1993); (g) Nakayama et al. (2010)

(h) Alfarra et al. (2006); (i) Lide (2008); (j) Riipinen et al. (2015) for Dp = 100 nm; (k) Gysel et al. (2007) for Dp = 60 nm;

(l) BC was assumed to be insoluble (e.g. Hung et al., 2015).

**Table2:**

| | Case I | Case II |
|---|---|---|
| RS: RH$_{ref}$ [%] | 60 | 40 |
| RL+MWR: RH$_{ref}$ [%] | 68 | 50 |
| RS: $f_\beta$(85%) | 1.50 | 2.60 |
| RL+MWR: $f_\beta$(85%) | 1.46 | 2.30 |
| $\gamma_{RS}$ | 0.56 ±0.01 | 0.99±0.01 |
| $\gamma_{RL+MWR}$ | 0.59 ±0.05 | 0.95±0.02 |

**Table 3:**

| | Measured | Calculated |
|---|---|---|
| $RH_{ref}$ [%] | 78 | 78 |
| $f_\beta^{532}$(85%) | 1.20 | 1.15 |
| $f_\beta^{355}$(85%) | 1.07 | 1.10 |
| $\gamma^{532}$ | 0.48 ±0.01 ($R^2$=0.84) | 0.53±0.02 ($R^2$=0.94) |
| $\gamma^{355}$ | 0.40 ±0.01($R^2$=0.65) | 0.45±0.02 ($R^2$=0.93) |

**Figure 1:**

[Figure]

**Figure 2**

[Figure]

**Figure 3:**
[Figure]

[Figure]

**Figure 4:**

[Figure]

**Figure 5:**

[Figure]

**Figure 6:**

[Figure]

**Figure 7:**